# On the Limitations of
# Stochastic Pre-processing Defenses

**Yue Gao**
University of Wisconsin–Madison
`gy@cs.wisc.edu`

**Ilia Shumailov**
University of Cambridge & Vector Institute
`ilia.shumailov@cl.cam.ac.uk`

**Kassem Fawaz**
University of Wisconsin–Madison
`kfawaz@wisc.edu`

**Nicolas Papernot**
University of Toronto & Vector Institute
`nicolas.papernot@utoronto.ca`

## Abstract

Defending against adversarial examples remains an open problem. A common belief is that randomness at inference increases the cost of finding adversarial inputs. An example of such a defense is to apply a random transformation to inputs prior to feeding them to the model. In this paper, we empirically and theoretically investigate such stochastic pre-processing defenses and demonstrate that they are flawed. First, we show that most stochastic defenses are weaker than previously thought; they lack sufficient randomness to withstand even standard attacks like projected gradient descent. This casts doubt on a long-held assumption that stochastic defenses invalidate attacks designed to evade deterministic defenses and force attackers to integrate the Expectation over Transformation (EOT) concept. Second, we show that stochastic defenses confront a trade-off between adversarial robustness and model invariance; they become less effective as the defended model acquires more invariance to their randomization. Future work will need to decouple these two effects. We also discuss implications and guidance for future research.

## 1 Introduction

Machine learning models are vulnerable to adversarial examples [4, 36], where an adversary can add imperceptible perturbations to the input of a model and change its prediction [5, 24]. Their discovery has motivated a wide variety of defense approaches [6, 8, 12, 29, 40, 42] along with the evaluation of their adversarial robustness [2, 28, 38]. Current evaluations mostly rely on adaptive attacks [2, 38], which require significant modeling and computational efforts. However, even when the attack succeeds, such evaluations may not always reveal the fundamental weaknesses of an examined defense. Without awareness of the underlying weaknesses, subsequent defenses may still conduct inadvertently weak adaptive attacks; this leads to overestimated robustness.

One popular class of defenses that demonstrates the above is the stochastic pre-processing defense, which relies on applying randomized transformations to inputs to provide robustness [12, 42]. Despite existing attack techniques designed to handle randomness [2, 3], there is an increasing effort to improve these defenses through a larger randomization space or more complicated transformations. For example, BaRT [28] employs 25 transformations, where the parameters of each transformation are further randomized. Due to the complexity of this defense, it was only broken recently (three years later) by Sitawarin et al. [34] with a complicated adaptive attack. Still, it is unclear how future defenses can avoid the pitfalls of existing defenses, largely because these pitfalls remain unknown.

36th Conference on Neural Information Processing Systems (NeurIPS 2022).

In this paper, we investigate stochastic pre-processing defenses and explain their limitations both empirically and theoretically. First, we revisit previous stochastic pre-processing defenses and explain why such defenses are broken. We show that most stochastic defenses are not sufficiently randomized to invalidate standard attacks designed for deterministic defenses. Second, we study recent stochastic defenses that exhibit more randomness and show that they also face key limitations. In particular, we identify a trade-off between their robustness and the model's invariance to their transformations. These defenses achieve a notion of robustness that results from reducing the model's invariance to the applied transformations. We outline our findings below. These findings suggest future work to find new ways of using randomness that decouples these two effects.

**Most stochastic defenses lack sufficient randomness.** While Athalye et al. [2] and Tramèr et al. [38] have demonstrated the ineffectiveness of several stochastic defenses with techniques like Expectation over Transformation (EOT) [3], it remains unclear whether and why EOT is required (or at least as a "standard technique") to break them. A commonly accepted explanation is that EOT computes the "correct gradients" of models with randomized components [2, 38], yet the necessity of such correct gradients has not been explicitly discussed. To fill this gap, we examine a long-held assumption that stochastic defenses invalidate standard attacks designed for deterministic defenses.

Specifically, we revisit stochastic pre-processing defenses previously broken by EOT and examine their robustness *without* applying EOT. Interestingly, we find that most stochastic defenses lack sufficient randomness to withstand even standard attacks (that do not integrate any strategy to capture model randomness) like projected gradient descent (PGD) [24]. We then conduct a systematic evaluation to show that applying EOT is only beneficial when the defense is sufficiently randomized. Otherwise, standard attacks already perform well and the randomization's robustness is overestimated.

**Trade-off between adversarial robustness and model invariance.** When stochastic pre-processing defenses do have sufficient randomness, they must fine-tune the model using augmented training data to preserve utility in the face of randomness added. We characterize this procedure by the model's *invariance* to the applied defense, where we identify a trade-off between the model's robustness (provided by the defense) and its invariance to the applied defense. Stochastic pre-processing defenses become less effective when their defended model acquires more invariance to their transformations.

On the theoretical front, we present a theoretical setting where this trade-off provably exists. We show from this trade-off that stochastic pre-processing defenses provide robustness by inducing variance on the defended model, and must take back such variance to recover utility. We verify this trade-off with empirical evaluations on realistic datasets, models, and defenses. We observe that robustness drops when the defended model is fine-tuned on data processed by its defense to acquire higher invariance.

## 2 Related Work

**Stochastic Pre-processing Defenses.** Defending against adversarial examples remains an open problem, where a common belief is that inference-time randomness increases the cost of finding adversarial inputs. Early examples of such stochastic defenses include input transformations [12] and rescaling [42]. These defenses were broken by Athalye et al. [2] using techniques like EOT [3] to capture randomness. After that, more stochastic defenses were proposed but with inadvertently weak evaluations [26, 29, 40, 44], which were found ineffective by Tramèr et al. [38]. Subsequent stochastic defenses resort to larger randomization space like BaRT [28], which was only broken recently by Sitawarin et al. [34]. In parallel to our work, DiffPure [25] adopts a complicated stochastic diffusion process to purify the inputs. As we will discuss in Appendix F.1, this defense belongs to an existing line of research that leverages generative models to pre-process input images [19, 32, 35], hence it matches the settings in our work. On the other hand, randomized smoothing [6, 17, 31] leverages randomness to certify the inherent robustness of a given decision. In this work, instead of designing adaptive attacks for individual defenses, which is a well-known challenging progress [2, 34, 38], we focus on the general stochastic pre-processing defenses and demonstrate their limitations.

**Trade-offs for Adversarial Robustness.** The trade-offs associated with adversarial robustness have been widely discussed in the literature. For example, prior work identified trade-offs between robustness and accuracy [39, 46] for deterministic classifiers. Pinot et al. [27] generalize this trade-off to randomized classifiers with a similar form as randomized smoothing. Compared with these results, our work provides a deeper understanding that stochastic pre-processing defenses *explicitly* control such trade-offs to provide robustness. Recent work also investigated the trade-off between the model's

robustness and invariance to input transformations, such as circular shifts [33] and rotations [14]. These trade-offs characterize a standalone model's own property — the model itself is less robust to adversarial examples when it becomes more invariant to certain transformations, without any defense. Our setting, however, is orthogonal to such analysis — the model that we consider is protected by a stochastic pre-processing defense, and what we really aim to characterize is the performance of that pre-processing defense, not the inherent robustness of the model itself.

## 3 Preliminaries

**Notations.**    Let $f : \mathcal{X} \to \mathbb{R}^C$ denote the classifier with pre-softmax outputs, where $\mathcal{X} = [0, 1]^d$ is the input space with $d$ dimensions and $C$ is the number of classes. We then consider a stochastic pre-processing defense $t_{\boldsymbol{\theta}} : \mathcal{X} \to \mathcal{X}$, where $\boldsymbol{\theta}$ is the random variable drawn from some randomization space $\Theta$ that parameterizes the defense. The defended classifier can be written as $f_{\boldsymbol{\theta}}(\boldsymbol{x}) := f(t_{\boldsymbol{\theta}}(\boldsymbol{x}))$.

Let $F(\boldsymbol{x}) := \arg\max_{i \in \mathcal{Y}} f_i(\boldsymbol{x})$ denote the classifier that returns the predicted label, where $f_i$ is the output of the $i$-th class and $\mathcal{Y} = [C]$ is the label space. Similarly, we use $F_{\boldsymbol{\theta}}$ and $f_{\boldsymbol{\theta},i}$ to denote the prediction and class-output of the stochastic classifier $f_{\boldsymbol{\theta}}$. Since this classifier returns varied outputs for a fixed input, it determines the final prediction by aggregating $n$ independent inferences with strategies like majority vote. We discuss these strategies and the choice of $n$ in Appendix A.1.

**Adversarial Examples.**    Given an image $\boldsymbol{x} \in \mathcal{X}$ and a classifier $F$, the adversarial example $\boldsymbol{x}' := \boldsymbol{x} + \boldsymbol{\delta}$ is visually similar to $\boldsymbol{x}$ but either misclassified (i.e., $F(\boldsymbol{x}') \neq F(\boldsymbol{x})$) or classified as a target class $y'$ chosen by the attacker (i.e., $F(\boldsymbol{x}') = y'$). Attack algorithms generate adversarial examples by searching for $\boldsymbol{\delta}$ such that $\boldsymbol{x}'$ fools the classifier while minimizing $\boldsymbol{\delta}$ under some distance metrics; for instance, the $\ell_p$ norm constraint $\|\boldsymbol{\delta}\|_p \leq \epsilon$ for a perturbation budget $\epsilon$.

**Projected Gradient Descent (PGD).**    PGD [24] is one of the most established attacks to evaluate adversarial example defenses. Given a benign example $\boldsymbol{x}^0$ and its ground-truth label $y$, each iteration of the untargeted PGD attack (with $\ell_\infty$ norm budget $\epsilon$) can be formulated as

$$\boldsymbol{x}^{i+1} \leftarrow \boldsymbol{x}^i + \alpha \cdot \text{sgn}\{\nabla \mathcal{L}(f_{\boldsymbol{\theta}}(\boldsymbol{x}^i), y)\}, \tag{1}$$

where $\alpha$ is the step size, $\mathcal{L}$ is the loss function, and each iteration is projected to the $\ell_\infty$ ball around $\boldsymbol{x}^0$ of radius $\epsilon$. We use PGD-$k$ to denote the PGD attack with $k$ steps. We outline formulations for other settings and norms in Appendix A.2.

**Expectation over Transformation (EOT).**    Since the classifier $f_{\boldsymbol{\theta}}$ is stochastic, the defense evaluation literature [2, 38] argues that attacks should target the *expectation* of the gradient using Expectation over Transformation (EOT) [3], which reformulates the PGD attack as

$$\boldsymbol{x}^{i+1} \leftarrow \boldsymbol{x}^i + \alpha \cdot \text{sgn}\Big\{\mathbb{E}_{\boldsymbol{\theta} \sim \Theta}\Big[\nabla \mathcal{L}(f_{\boldsymbol{\theta}}(\boldsymbol{x}^i), y)\Big]\Big\} \approx \boldsymbol{x}^i + \alpha \cdot \text{sgn}\Big\{\frac{1}{m}\sum_{j=1}^{m} \nabla \mathcal{L}(f_{\boldsymbol{\theta}_j}(\boldsymbol{x}^i), y)\Big\}, \tag{2}$$

where $m$ is the number of samples to estimate the expectation and $\boldsymbol{\theta}_j \overset{\text{iid}}{\sim} \Theta$ are sampled parameters for the defense. We use EOT-$m$ to denote the EOT technique with $m$ samples at each PGD step.

In addition, for a fair comparison among attacks with different PGD steps and EOT samples, we quantify the attack's strength by its total number of gradient computations. For example, attacks using PGD-$k$ and EOT-$m$ will have strength $k \times m$. Although white-box attacks are typically not constrained in this way, it allows for a fair comparison when attacks have finite computing resources (e.g., when EOT is not parallelizable). We discuss more about this quantification in Appendix A.3.

## 4 Most Stochastic Defenses Lack Sufficient Randomness

Athalye et al. [2] and Tramèr et al. [38] demonstrate adaptive evaluation of stochastic defenses with the application of EOT. However, it remains unclear why EOT is required (or at least as a "standard technique") to break these stochastic defenses. While a commonly accepted explanation is that EOT computes the "correct gradients" of models with randomized components [2, 38], the necessity of such correct gradients has not been explicitly discussed. To fill this gap, we revisit stochastic defenses previously broken by EOT and examine their robustness *without* applying EOT. Interestingly, we find that applying EOT is mostly *unnecessary* when evaluating existing stochastic defenses.

Table 2: The missing ablation study of adaptive evaluations of stochastic defenses in the literature. Notations: attack iterations $k$, EOT samples $m$, learning rate $\alpha$, number of gradient queries $k \times m$. The details of these defenses and their evaluation settings are in Appendix B.

| Defenses | Original Adaptive Evaluation (w/ EOT) | | | | | Our Ablation Study (w/o EOT) | | | | |
|---|---|---|---|---|---|---|---|---|---|---|
| | $k$ | $m$ | $\alpha$ | $k \times m$ | Success Rate | $k$ | $m$ | $\alpha$ | $k \times m$ | Success Rate |
| Guo et al. [12] | 1,000 | 30 | 0.1 | 30,000 | 100% | 1,000 | 1 | 0.001 | 1,000 | 99.0% |
| Xie et al. [42] | 1,000 | 30 | 0.1 | 30,000 | 100% | 200 | 1 | 0.1 | 200 | 100% |
| Dhillon et al. [8] | 500 | 10 | 0.1 | 5,000 | 100% | 500 | 1 | 0.1 | 500 | 100% |
| Xiao et al. [40] | 100 | 1,000 | 0.01 | 100,000 | 100% | 40,000 | 1 | 0.1/255 | 40,000 | 98.4% |
| Roth et al. [29] | 100 | 40 | 0.2/255 | 4,000 | 100% | 4,000 | 1 | 0.1/255 | 4,000 | 96.1% |

**Case Study: Random Rotation.** We start with a simple stochastic defense that randomly rotates the input image for $\theta \in [-90, 90]$ degrees (chosen at uniform) before classification. This defense is representative for most pre-processing defenses [12, 28, 42]. We evaluate this defense on 1,000 ImageNet images with PGD-$k$ and EOT-$m$ under the constraint $k \times m = 50$, as discussed in Section 3. All attacks use maximum $\ell_\infty$ perturbation $\epsilon = 8/255$ with step size chosen from $\alpha \in \{1/255, 2/255\}$. The results are shown in Table 1, where

Table 1: Evaluation of the random rotation with PGD-$k$ and EOT-$m$.

| Attacks | $k$ | $m$ | Success Rate |
|---|---|---|---|
| Untargeted | 10 | 5 | 100% |
| | 50 | 1 | 100% |
| Targeted | 10 | 5 | 99.0% |
| | 50 | 1 | 99.0% |

PGD-50 performs equally well as PGD-10 combined with EOT-5. This observation suggests that *some stochastic defenses are already breakable without applying EOT*, casting doubt on a long-held assumption that stochastic defenses simply invalidate attacks designed for deterministic defenses.

**Comprehensive Evaluations.** We then extend the above case study to other stochastic defenses evaluated in the literature. Specifically, we replicate the (untargeted) adaptive evaluation of stochastic defenses from Athalye et al. [2] and Tramèr et al. [38] with their official implementation. We only change the attack's hyper-parameters (e.g., number of iterations and learning rate) and disable EOT by setting its number of samples to one ($m = 1$), which avoids potential implementation flaws if removed from the source code. The comparison between evaluations with and without applying EOT is summarized in Table 2, which serves as a missing ablation study of adaptive evaluations in the literature. The experimental settings are identical within each row (detailed in Appendix B).

Interestingly, we find it *unnecessary* to break these defenses with EOT, as long as the standard attack runs for more iterations with a smaller learning rate. For such defenses, standard iterative attacks already contain an *implicit expectation* across iterations to capture the limited randomness. This observation implies that most stochastic defenses lack sufficient randomness to withstand even standard attacks designed for deterministic defenses. Therefore, increasing randomness becomes a promising approach to enhancing stochastic defenses, as adopted by recent defenses [6, 28]. Note that this ablation study only aims to inspire potential ways of enhancing stochastic defenses; it does not invalidate EOT for stronger adaptive evaluations of stochastic defenses.

## 5 Trade-offs between Robustness and Invariance

When stochastic pre-processing defenses *do have* sufficient randomness, they must ensure that the utility of the defended model is preserved in the face of randomness. To achieve high utility, existing defenses mostly rely on augmentation invariance through *trained invariance* [23]. In such a case, the invariance is achieved by applying the defense's randomness to the training data so as to guide the model in learning their transformations. For defenses based on stochastic pre-processor $t_\theta$, each data sample from the dataset gets augmented with $t_\theta$ sampled from the randomization space $\Theta$, and the risk is minimized over such augmented data.

The defended classifier $F_\theta(x) \coloneqq F(t_\theta(x))$ is invariant under the randomization space $\Theta$ if

$$F(t_\theta(x)) = F(x), \quad \forall \, \theta \in \Theta, x \in \mathcal{X}. \tag{3}$$

As we can observe from the definition, invariance has direct implications on the performance of stochastic pre-processing defenses. If the classifier is invariant under the defense's randomization space $\Theta$ as is defined in Equation (3), then the defense should not work – computing the model and its gradients over randomization $\theta \in \Theta$ is the same as if $t_\theta$ was not applied at all. This observation

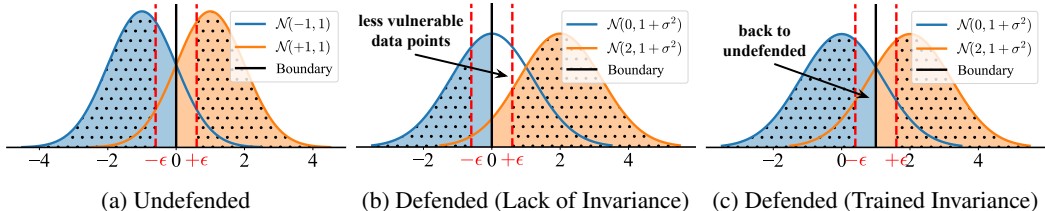

| (a) Undefended | (b) Defended (Lack of Invariance) | (c) Defended (Trained Invariance) |

Figure 1: Illustration of the binary classification task we consider. The curves are the probability density function of two classes of data. Shadowed area denotes correct classification. Dotted area denotes robustly correct classification under the $\ell_\infty$-bounded adversary with perturbation budget $\epsilon$.

suggests a direct coupling between invariance and performance of the defense: the more invariant, hence performant, the model is under a given randomization space, the less protection such a defense would provide. In this section, we present a simple theoretical setting where this coupling provably exists, as illustrated in Figure 1. Detailed arguments are deferred to Appendix C.1.

**Binary Classification Task.** We consider a class-balanced dataset $\mathcal{D}$ consisting of input-label pairs $(x, y)$ with $y \in \{-1, +1\}$ and $x|y \sim \mathcal{N}(y, 1)$, where $\mathcal{N}(\mu, \sigma^2)$ is a normal distribution with mean $\mu$ and variance $\sigma^2$. Moreover, an $\ell_\infty$-bounded adversary perturbs the input with a small $\delta$ to fool the classifier for $\|\delta\|_\infty \leq \epsilon$. We quantify the classifier's robustness by its robust accuracy, i.e., the ratio of correctly classified samples that remain correct after being perturbed by the adversary.

*Undefended Classification.* We start with the optimal linear classifier $F(x) := \text{sgn}(x)$ without any defense in Figure 1a. This classifier attains robust accuracy

$$\Pr\big[F(x+\delta) = y \mid F(x) = y\big] = \frac{\Pr\big[F(x+\delta) = y \wedge F(x) = y\big]}{\Pr\big[F(x) = y\big]} = \frac{\Phi(1-\epsilon)}{\Phi(1)}, \qquad (4)$$

where $\Phi$ is the cumulative distribution function of $\mathcal{N}(0, 1)$.

*Defended Classification.* We then try to improve adversarial robustness by introducing a stochastic pre-processing defense $t_\theta(x) := x + \theta$, where $\theta \sim \mathcal{N}(1, \sigma^2)$ is the random variable parameterizing the defense. This defense characterizes common pre-processing defenses that enforce randomness while shifting the input distribution. Here, the processed input follows a shifted distribution $t_\theta(x) \sim \mathcal{N}(y+1, 1+\sigma^2)$ in Figure 1b. The defended classifier $F_\theta(x) = \text{sgn}(x+\theta)$ has robust accuracy

$$\Pr\big[F_\theta(x+\delta) = y \mid F_\theta(x) = y\big] = \frac{\Pr\big[F_\theta(x+\delta) = y \wedge F_\theta(x) = y\big]}{\Pr\big[F_\theta(x) = y\big]} = \frac{\Phi'(-\epsilon) + \Phi'(2-\epsilon)}{\Phi'(0) + \Phi'(2)}, \quad (5)$$

where $\Phi'(x) := \Phi(x/\sqrt{1+\sigma^2})$ is the cumulative distribution function of $\mathcal{N}(0, 1+\sigma^2)$. At this point, we have not fit the classifier on processed inputs. Due to its lack of invariance, the defended classifier has low utility yet higher robust accuracy than the undefended one in Equation (4).

*Defended Classification (Trained Invariance).* As discussed above, one critical step of stochastic pre-processing defenses is to preserve the defended model's utility by minimizing the risk over augmented data $t_\theta(x)$, which leads to a new defended classifier $F_\theta^+(x) = \text{sgn}(x+\theta-1)$ in Figure 1c. As a result, this new defended classifier achieves higher invariance with robust accuracy

$$\Pr\big[F_\theta^+(x+\delta) = y \mid F_\theta^+(x) = y\big] = \frac{\Pr\big[F_\theta^+(x+\delta) = y \wedge F_\theta^+(x) = y\big]}{\Pr\big[F_\theta^+(x) = y\big]} = \frac{\Phi'(1-\epsilon)}{\Phi'(1)}, \quad (6)$$

which is less robust than the previous less-invariant classifier $F_\theta$ in Equation (5). However, one may observe that this classifier, though loses some robustness compared with $F_\theta$, is still more robust than the original undefended classifier $F$ in Equation (4). This part of robustness comes from the changed data distribution due to the defense's randomness. It shows that we have not achieved perfect invariance to the defense's randomness, thus gaining some robustness at the cost of utility.

*Defended Classification (Perfect Invariance).* Furthermore, these defenses usually leverage majority vote to obtain stable predictions, which finally produces a perfectly invariant defended classifier

$$F_\theta^*(x) = \text{sgn}\left\{\frac{1}{n}\sum_{i=1}^n F_{\theta_i}^+(x)\right\} = \text{sgn}\left\{\frac{1}{n}\sum_{i=1}^n \text{sgn}(x+\theta_i-1)\right\} \to \text{sgn}(x) = F(x), \qquad (7)$$

where $\theta_i \overset{\text{iid}}{\sim} \mathcal{N}(1, \sigma^2)$ are sampled parameters. In such a case, the defended classifier reduces to the original undefended classifier with the original robust accuracy:

$$\Pr\big[F_\theta^*(x + \delta) = y \mid F_\theta^*(x) = y\big] = \Pr\big[F(x + \delta) = y \mid F(x) = y\big] = \frac{\Phi(1 - \epsilon)}{\Phi(1)}. \tag{8}$$

*Summary.* The above theoretical setting illustrates how stochastic pre-processing defenses first induce variance on the binary classifier we consider to provide adversarial robustness in Equation (5), and how they finally take back such variance in Equations (6) and (8) to recover utility. We then extend the above coupling between robustness and invariance to a general trade-off in the following theorem, whose detailed descriptions and proofs are deferred to Appendices C.2 and C.3, respectively.

**Theorem 1** (Trade-off between Robustness and Invariance). *Given the above theoretical setting and assumptions, when the defended classifier $F_\theta(x)$ achieves higher invariance $R(k)$ under the defense's randomization space to preserve utility, the adversarial robustness provided by the defense strictly decreases.*

In a nutshell, we prove the strictly opposite monotonic behavior of robustness and invariance when the classifier shifts its decision boundary and employs majority vote to preserve utility. It shows that stochastic pre-processing defenses provide robustness by explicitly reducing the model's invariance to added randomized transformations, and the robustness disappears once the invariance is recovered.

# 6 Experiments

Our experiments are designed to answer the following two questions.

**Q1: What properties make applying EOT beneficial when evaluating stochastic defenses?**

We show that applying EOT is only beneficial when the defense is sufficiently randomized; otherwise standard attacks already perform well and leave no room for EOT to improve.

**Q2: What is the limitation of stochastic defenses when they do have sufficient randomness?**

We show a trade-off between the stochastic defense's robustness and the model's invariance to the defense itself. Such defenses become less effective when the defended model achieves higher invariance to their randomness, as required to preserve utility under the defense.

## 6.1 Experimental Settings

**Datasets & Models.** We conduct all experiments on ImageNet [30] and ImageNette [9]. For ImageNet, our test data consists of 1,000 images randomly sampled from the validation set. ImageNette is a ten-class subset of ImageNet, and we test on its validation set. We adopt various ResNet [13] models. For defenses with low randomness, we evaluate them on ImageNet with a pre-trained ResNet-50 with Top-1 accuracy 75.9%. For defenses with higher randomness (thus requiring fine-tuning), we switch to ImageNette and a pre-trained ResNet-34 with Top-1 accuracy 96.9% to reduce the training cost like previous work [34]. These models are fine-tuned on the training data processed by tested defenses. As a special case, we also evaluate randomized smoothing on ImageNet using the ResNet-50 models from Cohen et al. [6]. More details of datasets and models can be found in Appendices D.1 and D.2.

**Defenses & Metrics.** We focus on stochastic defenses allowing us to increase randomness: randomized smoothing [6] and BaRT [28]. For randomized smoothing, we vary the variance of the added Gaussian noise. For BaRT, we vary the number $\kappa$ of applied randomized transformations. Note that we have evaluated other stochastic defenses and discussed their low randomness in Section 4. We measure the defense's performance by the defended model's *benign accuracy* and the attack's *success rate*, all evaluated with majority vote over $n = 500$ predictions. The attack's success rate is the ratio of samples that do not satisfy the attack's objective prior to the attack but satisfy it after the attack. For example, we discard samples that were misclassified before being perturbed in untargeted attacks. Details of the evaluated defenses can be found in Appendix D.3.

**Attacks.** We evaluate defenses with standard PGD combined with EOT and focus on the $\ell_\infty$-bounded adversary with a perturbation budget $\epsilon = 8/255$ in both untargeted and targeted settings. We only use constant step sizes and no random restarts for PGD. We only conduct adaptive evaluations,

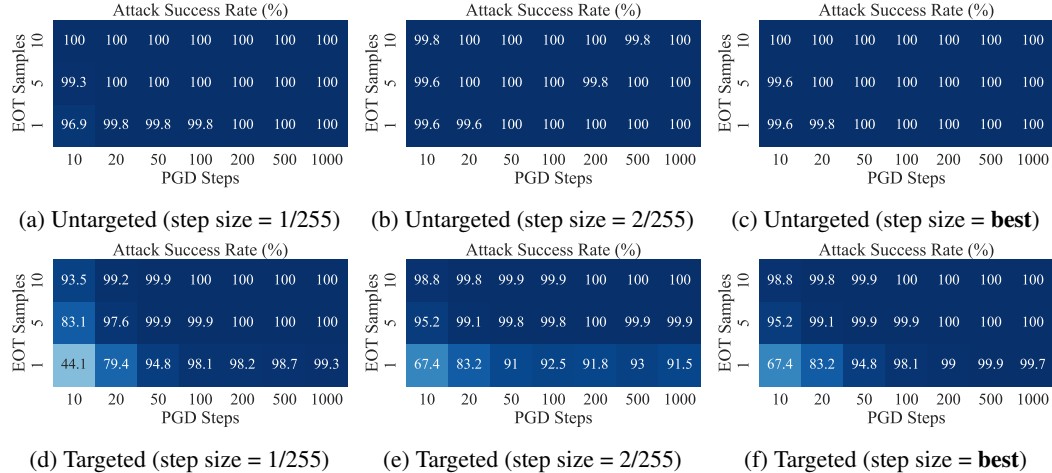

Figure 2: Evaluation of BaRT's noise injection defense on ImageNet. Standard PGD without applying EOT (i.e., applying EOT-1) is already good enough, leaving limited space for EOT to improve.

where the defense is included in the attack loop with non-differentiable components captured by BPDA [2]. We also utilize AutoPGD [7] to avoid selecting the step size when it is computationally expensive to repeat experiments in Section 6.3. More details and intuitions of the attack's settings and implementation can be found in Appendix D.4. Our code is available at https://github.com/wi-pi/stochastic-preprocessing-defenses.

## 6.2 Q1: Evaluate the Benefits of Applying EOT under Different Settings

In Section 4, we showed that standard attacks are sufficient to break most stochastic defenses due to their lack of randomness. Here, we aim to understand what properties make applying EOT beneficial when evaluating stochastic defenses. We design a systematic evaluation of stochastic defenses with different levels of randomness and check if applying EOT improves the attack.

**Stochastic Defenses with Low Randomness.** We start with BaRT's noise injection defense, which perturbs the input image with noise of distributions and parameters chosen at random. While this defense has low randomness, it yields meaningful results. We evaluate this defense with various combinations of PGD and EOT[1]. The performance of untargeted and targeted attacks is shown in Figure 2. We test multiple step sizes and summarize their best results (discussed in Appendix E.1).

In this case, standard PGD attacks are already good enough when the defense has insufficient randomness, leaving no space for improvements from EOT. In Figure 2f, both (1) PGD-10 combined with EOT-10 and (2) PGD-100 combined with EOT-1 have near 100% success rates. This result is consistent with our observations in Section 4 in both untargeted and targeted settings[2].

**Stochastic Defenses with Higher Randomness.** We then examine the randomized smoothing defense that adds Gaussian noise to the input image. Although this defense was originally proposed for certifiable adversarial robustness, we adopt it to evaluate how randomness affects the benefits of applying EOT. Similarly, we evaluate this defense with PGD and EOT of different settings with a focus on the *targeted* attack. The results are shown in Figure 3.

We observe that EOT starts to improve the attack when the defense has a higher level of randomness. For a fixed number of PGD steps, applying EOT significantly improves the attack in most of the settings. For a fixed attack strength (i.e., number of gradient computations), applying EOT always outperforms standalone PGD. In Figure 3f, for example, PGD-100 combined with EOT-10 is 5.5% higher than PGD-1,000 with EOT-1 (40.3% vs. 34.8%).

---

[1]Note that we do not intend to find a heuristic for the best combination of PGD-$k$ and EOT-$m$, as it is out of the scope of the question that we want to answer. However, it is still possible to correlate the choice of $k$ and $m$ with the convergence analysis of stochastic gradient descent, which we will briefly discuss in Appendix A.3.

[2]The only caveat is that targeted attacks are more likely to benefit from EOT, as their objectives are stricter and may have better performance with gradients of higher precision.

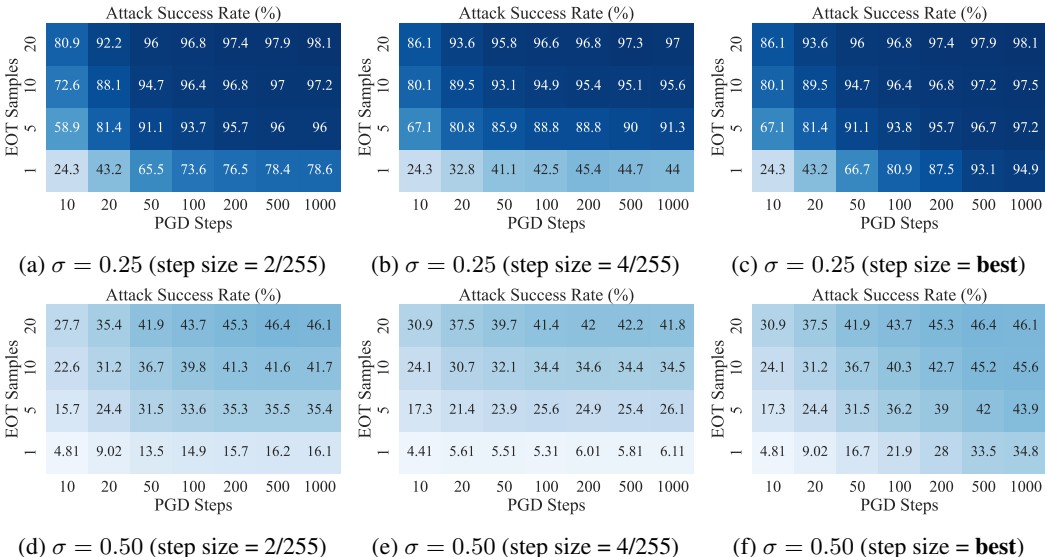

(a) $\sigma = 0.25$ (step size = 2/255) (b) $\sigma = 0.25$ (step size = 4/255) (c) $\sigma = 0.25$ (step size = **best**)

(d) $\sigma = 0.50$ (step size = 2/255) (e) $\sigma = 0.50$ (step size = 4/255) (f) $\sigma = 0.50$ (step size = **best**)

Figure 3: Evaluation of randomized smoothing on ImageNet (targeted attacks). PGD performs well on lower variance ($\sigma = 0.25$) if running for more steps. For a larger variance ($\sigma = 0.50$), applying EOT starts to improve the attack significantly (for a fixed number of gradient computations).

**Takeaways.** Applying EOT is only beneficial when the defense has sufficient randomness, such as randomized smoothing with $\sigma = 0.5$. This observation suggests that stochastic defenses only make standard attacks suboptimal when they have sufficient randomness. However, most existing stochastic defenses did not achieve this criterion, as we showed in Section 4. We also provide visualizations of adversarial examples under different settings and CIFAR10 results in Appendices E.3 and E.4.

### 6.3 Q2: Evaluate the Trade-off between Robustness and Invariance

In Section 5, we present a theoretical setting where the trade-off between robustness and invariance provably exists; stochastic defenses become less robust when the defended model achieves higher invariance to their randomness. Here, we demonstrate this trade-off on realistic datasets, models, and defenses. In particular, we choose defenses with sufficient randomness (achieved in different ways) and compare their performance when being applied to models of different levels of invariance, where the invariance is achieved by applying the defense's randomness to the training data so as to guide the model in learning their transformations.

**Randomness through Transformations.** We first examine the BaRT defense, which pre-processes input images with $\kappa$ randomly composited stochastic transformations. It represents defenses aiming to increase randomness through diverse input transformations. Since our objective is to demonstrate the trade-off, it suffices to evaluate a subset of BaRT with $\kappa \leq 6$ transformations; this also avoids the training cost of evaluating the original BaRT with $\kappa = 25$. Figure 4 shows the performance of this defense with models before and after fine-tuning on its processed training data.

In Figures 4a and 4c, we first observe that fine-tuning indeed increases the model's invariance to the applied defense's randomness; the utility's dashed green curves are improved to the solid green curves beyond 90%. However, as the model achieves higher invariance, the defense becomes nearly ineffective; the attack's dashed red curves boost to the solid red curves near 100%. The same attack's effectiveness throughout the fine-tuning procedure further verifies this observation, as shown in Figures 4b and 4d. It shows a clear trade-off between the defense's robustness and the model's invariance. That is, stochastic defenses start to lose robustness when their defended models achieve higher invariance to their transformations.

**Randomness through Noise Levels.** We then examine the randomized smoothing defense that adds Gaussian noise to the input image. Unlike BaRT's diverse transformations, randomized smoothing increases randomness directly through the added noise's variance $\sigma^2$. This allows us to rigorously increase the randomness without unexpected artifacts like non-differentiable components. We evaluate

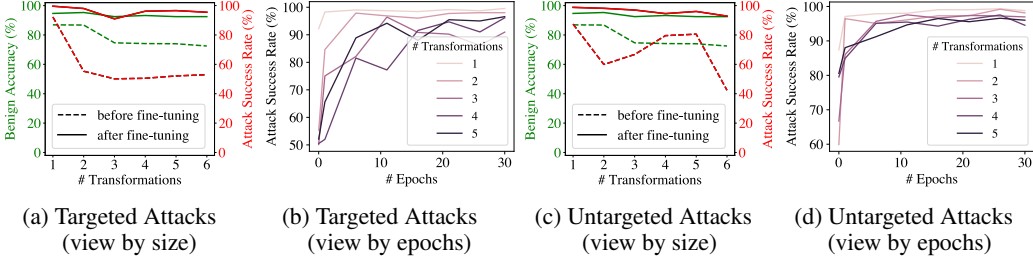

(a) Targeted Attacks (view by size)  (b) Targeted Attacks (view by epochs)  (c) Untargeted Attacks (view by size)  (d) Untargeted Attacks (view by epochs)

Figure 4: Performance of the BaRT defense on ImageNette with different numbers of transformations before and after fine-tuning the model. While the model achieves higher invariance, the defense becomes nearly ineffective[3], as evident from the top solid red curves in (a) and (c).

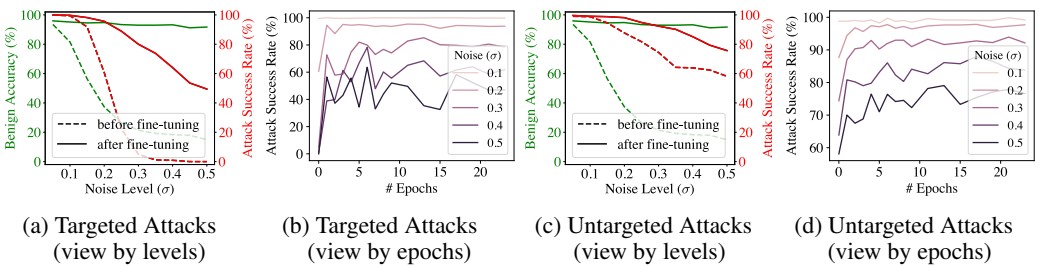

(a) Targeted Attacks (view by levels)  (b) Targeted Attacks (view by epochs)  (c) Untargeted Attacks (view by levels)  (d) Untargeted Attacks (view by epochs)

Figure 5: Performance of the randomized smoothing defense on ImageNette with different noise levels before and after fine-tuning the model. While the model achieves higher invariance, the defense becomes less effective[4], as evident from the gap between dashed and solid red curves in (a) and (c).

the performance of this defense ($\sigma \leq 0.5$) with models before and after fine-tuning on training data perturbed with designated Gaussian noise. The results are shown in Figure 5.

In Figures 5a and 5c, fine-tuning improves the model's invariance, but the defense also becomes significantly weaker during this process. For example, the targeted attack is nearly infeasible when the model is variant to the large noise ($\sigma \geq 0.3$), yet is significantly more effective when the model becomes invariant. The fine-tuning process in Figures 5b and 5d also verifies that stochastic defenses become weaker when their defended models become more invariant to their randomness.

**Takeaways.** For both the BaRT and the randomized smoothing defense, we observe a clear trade-off between the defense's robustness and the model's invariance to randomness, especially in the targeted setting. In particular, we find that stochastic defenses lose adversarial robustness when their defended models achieve higher invariance to their randomness. Our finding implies that such defenses would become ineffective when their defended models are perfectly invariant to their randomness.

## 7 Discussions

In this section, we discuss several questions that arose from our study of stochastic pre-processing defenses. Discussions about extensions, limitations, and broader topics can be found in Appendix F.

**What do stochastic pre-processing defenses really do?** We show that stochastic pre-processing defenses do not introduce inherent robustness to the prediction task. Instead, they shift the input distribution through randomness and transformations, which results in variance and introduces errors during prediction. The observed "robustness", in an unusual meaning for this literature, is a result of these errors. This is fundamentally different from the inherent robustness provided by adversarial training [24]. Although defenses like adversarial training still cost accuracy [39, 46], they do not intentionally introduce errors like stochastic pre-processing defenses.

---

[3]The defense may not grow stronger with more transformations, which is a drawback of BaRT that we will discuss in Appendix D.3. Yet, our evaluations focus on the fact that solid curves are above the dashed curves.

[4]One may also observe a trade-off between robustness and *utility* by examining the curve's horizontal trend. However, we focus on the trade-off between robustness and *invariance*, which manifests in the vertical gap.

**What are the concrete settings that stochastic pre-processing defenses work?** These defenses *do make* the attack harder when the adversary has only limited knowledge of the defense's transformations, e.g., in a low-query setting. In such a case, the defense practically introduces noise to the attack's optimization procedure, making it difficult for a low-query adversary to find adversarial examples that consistently cross the probabilistic decision boundary. However, it is still possible for the adversary to infer pre-processors in a black-box model and compute their expectation locally [10, 41], unless the randomization space changes over time. Our theoretical analysis considers a powerful adversary with full knowledge of the defense's randomization space; hence it can optimize directly towards the defended model's decision boundary in expectation. The other setting is randomized smoothing, which remains effective in certifying the inherent robustness of a given decision.

**What are the implications for future research?** Our work suggests that future defenses should try to decouple robustness and invariance; that is, either avoid providing robustness by introducing variance to the added randomness or the variance only applies to adversarial inputs. This implication is crucial as the research community continues improving defenses through more complicated transformations. For example, in parallel to our work, DiffPure [25] adopts a complicated stochastic diffusion process to purify the inputs. However, fully understanding DiffPure's robustness requires substantial effort due to its complications and high computational costs, as we will discuss in Appendix F.1

**How should we improve stochastic defenses?** Stochastic defenses should rely on randomness that exploits the properties of the prediction task. One promising approach is dividing the problem into orthogonal subproblems. For example, some speech problems like keyword spotting are inherently divisible in the spectrum space [1], and vision tasks are divisible by introducing different modalities [43], independency [18], or orthogonality [45]. In such cases, randomization forces the attack to target all possible (independent) subproblems, where the model performs well on each (independent and) non-transferable subproblem. As a result, defenses can decouple robustness and invariance, hence reducing the effective attack budget and avoiding the pitfall of previous randomized defenses. While systematic guidance for designing defenses (and their attacks) remains an open question, we summarize some critical insights along this direction in Appendix F.2.

**What are the implications for adaptive attackers?** Our findings suggest that an adaptive attacker needs to consider the spectrum of available standard attack algorithms, instead of just focusing on a given attack algorithm because of the defense's design. As we discover in this paper, EOT can be unnecessary for seemingly immune stochastic defenses, yet its application to break these said defenses gives a false impression about their security against weak attackers. When evaluating the robustness of a defense, the adaptive attack should start by tuning standard approaches, before resorting to more involved attack strategies. This approach helps us to identify the minimally capable attack that breaks the defense and develop a better understanding of the defense's fundamental weaknesses.

## 8 Conclusion

In this paper, we investigate stochastic pre-processing defenses and explain their limitations both empirically and theoretically. We show that most stochastic pre-processing defenses are weaker than previously thought, and recent defenses that indeed exhibit more randomness still face a trade-off between their robustness and the model's invariance to their transformations. While defending against adversarial examples remains an open problem and designing proper adaptive evaluations is arguably challenging, we demonstrate that stochastic pre-processing defenses are fundamentally flawed in their current form. Our findings suggest that future work will need to find new ways of using randomness that decouples robustness and invariance.

## Acknowledgement

We thank all anonymous reviewers for their insightful comments and feedback. We would like to acknowledge our sponsors, who support our research with financial and in-kind contributions: the DARPA GARD program under agreement number 885000, NSF through award CNS-2003129, CIFAR through the Canada CIFAR AI Chair program, and NSERC through the Discovery Grant and COHESA Strategic Alliance. Resources used in preparing this research were provided, in part, by the Province of Ontario, the Government of Canada through CIFAR, and companies sponsoring the Vector Institute. We would like to thank members of the CleverHans Lab for their feedback.

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
