**Appendix: On the Limitations of Stochastic Pre-processing Defenses**

**Table of Contents**

# A  More Preliminaries

In this section, we expand on the preliminaries provided in Section 3.

## A.1  Aggregation Strategies for Stochastic Classifiers

As the stochastic classifier $f_{\boldsymbol{\theta}}$ returns varied outputs even for a fixed input, it needs to determine the final prediction by aggregating $n$ independent inferences with a particular strategy.

**Majority Vote.** The most commonly used strategy is *majority vote*, which can be formulated as

$$F_{\boldsymbol{\theta}}^{\text{vote}}(\boldsymbol{x}) := \arg\max_{y \in \mathcal{Y}} \sum_{i=1}^{n} \mathbb{1}\left\{ F_{\boldsymbol{\theta}_i}(\boldsymbol{x}) = y \right\}, \tag{9}$$

where $\{\boldsymbol{\theta}_i\}_{i=1}^{n} \overset{\text{iid}}{\sim} \Theta$ are sampled parameters. We adopt this strategy when the defended classifier computes its prediction. But when attacking a defended classifier, we use the gradient obtained from the original stochastic classifier $f_{\boldsymbol{\theta}}$.

**Match All.** A more restricted strategy is *match all*, which requires all predictions to be identical:

$$F_{\boldsymbol{\theta}}^{\text{all}}(\boldsymbol{x}) := y, \quad \text{s.t.} \quad \sum_{i=1}^{n} \mathbb{1}\left\{ F_{\boldsymbol{\theta}_i}(\boldsymbol{x}) = y \right\} = n, \tag{10}$$

where $\{\boldsymbol{\theta}_i\}_{i=1}^{n} \overset{\text{iid}}{\sim} \Theta$ are sampled parameters. This strategy *rejects* the input if the condition is not satisfied, which can be used as a strict setting for targeted attacks. We do not choose this strategy in our work because it is overly strict and is hard to satisfy, even for benign inputs.

**Averaged Logits.** One may also determine the label from averaged logits over multiple inferences:

$$F_{\boldsymbol{\theta}}^{\text{logits}}(\boldsymbol{x}) := \arg\max_{y \in \mathcal{Y}} \frac{1}{n} \sum_{i=1}^{n} f_{\boldsymbol{\theta}_i, y}(\boldsymbol{x}), \tag{11}$$

where $\{\boldsymbol{\theta}_i\}_{i=1}^{n} \overset{\text{iid}}{\sim} \Theta$ are sampled parameters. Sitawarin et al. [34] leverage this strategy to design adaptive attacks against the BaRT [28] defense. Still, we do not use this strategy because our main objective is not to break defenses but to analyze their fundamental weaknesses. We only apply the prediction strategy when using a stochastic classifier to evaluate a given set of inputs.

**The Choice of Prediction Times.** The choice of $n$ typically depends on the stochastic classifier's variance to its randomness. In our setting, the randomness comes from the applied pre-processing defense $t_{\boldsymbol{\theta}}$ with random variable $\boldsymbol{\theta}$ drawn from the randomization space $\Theta$. When the randomization space is small, it suffices to set $n = 1$ for most such defenses [12, 42]. For defenses with a slightly larger randomization space, they can set $n = 30$, for example for randomized cropping [12]. Finally, defenses with even larger randomization spaces set $n = 500$ or more [6, 28]. We fix $n = 500$ in our main experiments in Section 6 for consistency.

## A.2  Formulation of Projected Gradient Descent

In this paper, we mainly use PGD [24] to evaluate the robustness of a stochastically defended model. Given a benign example $\boldsymbol{x}^0$ and its ground-truth label $y$, each iteration of the *untargeted* PGD attack with $\ell_{\infty}$ norm budget $\epsilon$ can be formulated as

$$\boldsymbol{x}^{i+1} \leftarrow \boldsymbol{x}^i + \alpha \cdot \text{sgn}\left\{ \nabla \mathcal{L}\left( f_{\boldsymbol{\theta}}(\boldsymbol{x}^i), y \right) \right\}, \tag{12}$$

where $\alpha$ is the step size, $\mathcal{L}$ is the loss function, and each iteration is projected to the $\ell_{\infty}$ ball around $\boldsymbol{x}^0$ of radius $\epsilon$. As for *targeted* PGD attacks with a target label $y'$, the above iteration becomes

$$\boldsymbol{x}^{i+1} \leftarrow \boldsymbol{x}^i - \alpha \cdot \text{sgn}\left\{ \nabla \mathcal{L}\left( f_{\boldsymbol{\theta}}(\boldsymbol{x}^i), y' \right) \right\}, \tag{13}$$

where we switch the optimizing direction and the label for computing the loss.

Similarly, the untargeted attack with $\ell_2$ norm budget $\epsilon$ is formulated as

$$\boldsymbol{x}^{i+1} \leftarrow \boldsymbol{x}^i + \alpha \cdot \left\| \nabla \mathcal{L}\left( f_{\boldsymbol{\theta}}(\boldsymbol{x}^i), y \right) \right\|_2, \tag{14}$$

where each iteration is projected to the $\ell_2$ norm ball around $\boldsymbol{x}^0$ of radius $\epsilon$, and the targeted attack

$$\boldsymbol{x}^{i+1} \leftarrow \boldsymbol{x}^i - \alpha \cdot \left\| \nabla \mathcal{L}\left( f_{\boldsymbol{\theta}}(\boldsymbol{x}^i), y' \right) \right\|_2. \tag{15}$$

### A.3 Quantifying the Strength of White-box Attacks

In this work, we consider attacks with different combinations of PGD steps and EOT samples, denoted by PGD-$k$ and EOT-$m$. For evaluations of deterministic defenses, quantifying the strength of PGD attacks by the number of steps $k$ is valid. However, this quantification is not informative enough when the evaluated defense is stochastic and involves EOT. For example, it is hard to tell whether PGD-1 with EOT-100 or PGD-100 with EOT-1 has more strength in terms of the number of steps. For a fair comparison between such attacks, we quantify their strength by the total *number of gradient computations*, defined as

$$\mathtt{strength}(\text{PGD-}k, \text{EOT-}m) := k \times m. \tag{16}$$

This concept is similar to the *query budget* in the black-box setting. Although we *do not* constrain white-box attacks like this, it allows for a fair comparison between attacks with different settings. For example, we can now argue that the two attacks above have the same strength due to $k \times m = 100$.

Moreover, the above quantification has realistic implications for the white-box attack's computational cost under finite computing resources (w.r.t. the number of evaluated samples). In such a case, the computation of EOT is not parallelizable by batching the EOT samples. For example, when attacking $B$ samples with a maximally possible batch size of $B$, the attacker has to compute the gradients for $k \times m$ batches. Only when the maximally possible batch size becomes $m \times B$, the attacker can parallelize the EOT samples and only needs to compute gradients for $k$ batches.

**Potential Optimality Analysis.** Although we evaluate various combinations of PGD-$k$ and EOT-$m$, we are not interested in finding a heuristic for the best combination for two reasons. Firstly, this discussion is beyond the scope of the question that we want to answer in Section 6.2. Secondly, white-box attackers in the real world have sufficient incentive to adopt a sufficiently large value of $k$ and $m$ (to make sure their attack converges), regardless of the potential optimal choice.

However, it is still possible to correlate the choice of $k$ and $m$ with the convergence rate of stochastic gradient descent (SGD). For example, it is well-known that the convergence rate of SGD can be affected by the estimated gradient's variance [11], and this variance is again affected by the number of EOT samples $m$ we choose due to the central limit theorem. As a result, one can analyze the attack's convergence behavior with different choices of PGD-$k$ and EOT-$m$. Still, this discussion is beyond the scope of this work and is more beneficial in the context of black-box attacks.

## B  Experiment Setup: Most Stochastic Defenses Lack Sufficient Randomness

In Section 4, we replicate the evaluation of five previous stochastic defenses from Athalye et al. [2][5] and Tramèr et al. [38][6] without applying EOT. Here, we provide more details of these defenses and their evaluation settings.

**Case Study: Random Rotation.** In this case study, we evaluate this defense on 1,000 randomly chosen ImageNet images and a pre-trained ResNet-50 model. The settings are consistent with our main evaluation described later in Appendix D.

### B.1  Randomized Image Cropping [12]

**Defense Details.** This defense randomly crops $m = 30$ patches of size $90 \times 90$ from each input image of size $299 \times 299$. These patches are sent to the classifier, and the final prediction is a majority vote over the predictions of these patches.

**Original Evaluation.** Athalye et al. [2] evaluate this defense with an $\ell_2$-bounded adversary under the root-mean-square perturbation budget of 0.05. Their attack decreases the classification loss (averaged over $m$ patches) using gradient descent with 1,000 iterations and a learning rate 0.1. They decrease the accuracy of an Inception-v3 [37] target model to 0% (i.e., 100% attack success rate) on 1,000 randomly sampled ImageNet [30] images.

**Our Ablation Study.** We replicate this evaluation by setting $m = 1$ when running the attack (the final defense still uses $m = 30$ patches). This means that we only attack a randomly cropped small

---

[5]https://github.com/anishathalye/obfuscated-gradients
[6]https://github.com/wielandbrendel/adaptive_attacks_paper

patch from the entire image at each iteration. We then change the learning rate from 0.1 to 0.001 and are able to achieve 99.0% attack success rate.

## B.2 Randomized Image Rescaling [42]

**Defense Details.** This defense randomly rescales the input image of size $299 \times 299$ to $r \times r$, where $r \in [200, 331)$ is chosen uniformly at random, and then randomly pads the image with zeros to size $331 \times 331$. The resulting padded image is sent to the classifier for one-time prediction.

**Original Evaluation.** Athalye et al. [2] evaluate this defense with an $\ell_\infty$-bounded adversary under the perturbation budget of 8/255. They generate adversarial examples using PGD-1000 with a step size of 0.1, where each step applies EOT-30 to compute the gradients averaged over 30 samples processed from the evaluated input image. They decrease the accuracy of an Inception-v3 [37] target model to 0% (i.e., 100% attack success rate) on 1,000 randomly sampled ImageNet [30] images.

**Our Ablation Study.** We replicate this evaluation with PGD-200 and EOT-1, with all the other parameters unchanged. We are still able to achieve a 100% attack success rate in this case.

## B.3 Randomized Activation Pruning [8]

**Defense Details.** This defense randomly drops (zeros out) some neurons of each layer with probability proportional to their absolute value. The defense considers several levels of probability, and we use the setting used by Athalye et al. [2].

**Original Evaluation.** Athalye et al. [2] evaluate this defense with an $\ell_\infty$-bounded adversary under the perturbation budget of 8/255. They decrease the margin between the correct label's logit and the wrong label's logit with gradient descent using the Adam [15] optimizer. The attack runs for 500 steps with a learning rate 0.1, where each iteration averages the gradient over 10 samples. The attack achieves 100% success rate on an Inception-v3 [37] target model and the CIFAR-10 [16] dataset.

**Our Ablation Study.** We replicate this evaluation by simply setting the number of EOT samples to 1 and are still able to obtain 100% success rate.

## B.4 Discontinuous Activation [40]

**Defense Details.** This defense replaces the standard ReLU activation function inside the neural network with a discontinuous function, so that only the $k$ largest elements are preserved. Although this defense is not stochastic by itself, we evaluate it because the existing evaluation relies heavily on the application of EOT.

**Original Evaluation.** Tramèr et al. [38] evaluate this defense with several techniques that approximate the correct gradient. For each input, they estimate the average local gradient with $m = 20,000$ random perturbations drawn from a standard normal distribution with standard deviation $\epsilon = 8/255$. Given this estimated gradient, they consider an $\ell_\infty$-bounded adversary with perturbation budget 8/255 and run the PGD attack with 100 steps with step size 0.01. Their evaluation code uses a fine-grained choice of $m$, which is set to 100, 1K, and 20K at the 1st, 20th, and 40th iterations, respectively. We report 1K in the main paper.

As a result, their attack achieves 100% attack success rate on a ResNet-18 [40] model from the original defense and the CIFAR-10 [16] dataset.

**Our Ablation Study.** We replicate this evaluation by moving all gradient computations from the estimation side $m$ to the attack's iteration side $k$. That is, instead of running PGD-100 with EOT-1K (100K gradient computations), we run PGD-40K and EOT-1 (40K gradient computations). This setting achieves 98.4% success rate on the same model and dataset. In Appendix E.1, we discuss an interesting observation when evaluating this defense; it shows that PGD may capture randomness as well as EOT with a carefully fine-tuned learning rate.

## B.5 Statistical Detection [29]

**Defense Details.** This defense is a statistical test for detecting adversarial examples. It checks if a given input image is overly robust under Gaussian noise, which is a property of adversarial examples generated by PGD [24] and C&W [5]; benign images are sensitive to such noise.

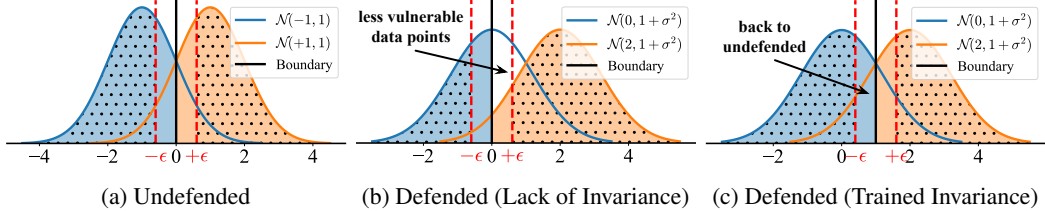

|   | (a) Undefended | (b) Defended (Lack of Invariance) | (c) Defended (Trained Invariance) |

Figure 6: Illustration of the binary classification task we consider. The curves are the probability density function of two classes of data. Shadowed area denotes correct classification. Dotted area denotes robustly correct classification under the $\ell_\infty$-bounded adversary with perturbation budget $\epsilon$.

**Original Evaluation.** Tramèr et al. [38] evaluate this defense with logit matching. Specifically, they generate adversarial examples with a logit that matches a given target image's logit in terms of (1) low mean squared error (MSE) distance and (2) similar robustness under the Gaussian noise. Their attack combines the above two objectives and runs for 100 steps with a learning rate 0.2/255, where the robustness under the Gaussian noise is measured under $m = 40$ samples. The resulting PGD-100 and EOT-40 attack achieves 100% success rate on a target ResNet [13] model and 1,000 randomly sampled ImageNet [30] images.

**Our Ablation Study.** We replicate the evaluation by moving all EOT samples to PGD steps. That is, we run PGD-4K and EOT-1 with a learning rate 0.1/255. As a result, our attack achieves 96.1% success rate, only 3.9% lower than the attack using EOT. We did not tune the step size further.

## C   Theoretical Analysis: Trade-off between Robustness and Invariance

We consider a class-balanced dataset $\mathcal{D}$ consisting of input-label pairs $(x, y)$ with $y \in \{-1, +1\}$ and $x|y \sim \mathcal{N}(y, 1)$, where $\mathcal{N}(\mu, \sigma^2)$ is a normal distribution with mean $\mu$ and variance $\sigma^2$. An $\ell_\infty$-bounded adversary perturbs the input with a small $\delta$ to fool the classifier for $\|\delta\|_\infty \le \epsilon$. We quantify the classifier's robustness by its robust accuracy, i.e., the ratio of correctly classified samples that remain correct after being perturbed by the adversary. We also consider a stochastic pre-processing defense $t_\theta(x) := x + \theta$, where $\theta \sim \mathcal{N}(1, \sigma^2)$ is the random variable parameterizing the defense.

We formalize our assumptions as follows. Assumptions 1 to 3 characterize the standard behavior of classifiers that employ the pre-processing defense, and Assumption 4 specifies a set of hyper-parameters to simplify the analysis without loss of generality.

**Assumption 1** (pre-processing defense). *The classifier only employs a pre-processing defense of the form $t_\theta(x) := x + \theta$. As such, the defended classifier is defined as $F_\theta(x) := \operatorname{sgn}(x + \theta - k)$, where $k$ is the decision boundary it wants to optimize.*

**Assumption 2** (trained invariance). *The defended classifier controls its invariance to the defense's transformation through trained invariance, i.e., shifting the decision boundary $k$.*

**Assumption 3** (majority vote). *The defended classifier employs majority vote (for higher invariance) only after it improves the trained invariance. We only consider a sufficiently large number of votes.*

**Assumption 4** (hyper-parameters). *For simplicity, we assume that the defender applies $\theta \sim \mathcal{N}(1, 1)$ and the adversary is reasonably strong with a perturbation budget $\epsilon = 1$. Note that $\epsilon = 1$ allows the adversary to shift half of the data across the decision boundary in the undefended scenario.*

**Disambiguation of Notations.** We use $x + \delta$ to denote the perturbed input passed to a classifier, such as $F(x + \delta)$, but its actual value can be any value chosen from $[x - \epsilon, x + \epsilon]$. We use $\varphi$ and $\Phi$ to denote the PDF and CDF of the standard normal distribution $\mathcal{N}(0, 1)$, respectively. We use $\varphi'$ and $\Phi'$ to denote the PDF and CDF for non-standard normal distributions, whose parameters will be specified in the context.

### C.1   Detailed Analysis of the Binary Classification Task

We outline the detailed computations of Section 5 below.

### C.1.1 Undefended Classification

The Bayes optimal linear classifier $F(x) = \text{sgn}(x)$ without any defense is illustrated in Figure 6a. This classifier has benign accuracy (i.e., shadowed area):

$$
\begin{aligned}
\Pr[F(x) = y] &= \frac{1}{2}\Big(\Pr[F(x) = y \mid y = -1] + \Pr[F(x) = y \mid y = +1]\Big) \\
&= \frac{1}{2}\Big(\Pr[x < 0 \mid y = -1] + \Pr[x > 0 \mid y = +1]\Big) \\
&= \frac{1}{2}\Big(\Pr[\mathcal{N}(-1,1) < 0] + \Pr[\mathcal{N}(+1,1) > 0]\Big) \\
&= \frac{1}{2}\Big(\Phi(1) + 1 - \Phi(-1)\Big) \\
&= \Phi(1).
\end{aligned}
\tag{17}
$$

We then compute its probability of making robustly correct predictions (i.e., dotted area)

$$
\begin{aligned}
&\Pr[F(x+\delta) = y \wedge F(x) = y] \\
&= \frac{1}{2}\Big(\Pr[F(x+\delta) = y \wedge F(x) = y \mid y = -1] + \Pr[F(x+\delta) = y \wedge F(x) = y \mid y = +1]\Big) \\
&= \frac{1}{2}\Big(\Pr[x + \epsilon < 0 \wedge x < 0 \mid y = -1] + \Pr[x - \epsilon > 0 \wedge x > 0 \mid y = +1]\Big) \\
&\quad \text{(where we use } x - \epsilon \text{ when } y = +1 \text{ because the correctly classified sample must lie on the right)} \\
&= \frac{1}{2}\Big(\Pr[x < -\epsilon \mid y = -1] + \Pr[x > \epsilon \mid y = +1]\Big) \\
&= \frac{1}{2}\Big(\Pr[\mathcal{N}(-1,1) < -\epsilon] + \Pr[\mathcal{N}(+1,1) > \epsilon]\Big) \\
&= \frac{1}{2}\Big(\Phi(1 - \epsilon) + 1 - \Phi(\epsilon - 1)\Big) \\
&= \Phi(1 - \epsilon),
\end{aligned}
\tag{18}
$$

which shows that this classifier has robust accuracy (i.e., dotted area over shadowed area)

$$
\Pr[F(x+\delta) = y \mid F(x) = y] = \frac{\Pr[F(x+\delta) = y \wedge F(x) = y]}{F(x) = y} = \frac{\Phi(1 - \epsilon)}{\Phi(1)},
\tag{19}
$$

which verifies the computation in Equation (4).

### C.1.2 Defended Classification

The defended classifier $F_\theta(x) = \text{sgn}(x + \theta)$ is illustrated in Figure 6b, with benign accuracy

$$
\begin{aligned}
\Pr[F_\theta(x) = y] &= \frac{1}{2}\Big(\Pr[F_\theta(x) = y \mid y = -1] + \Pr[F_\theta(x) = y \mid y = +1]\Big) \\
&= \frac{1}{2}\Big(\Pr[x + \theta < 0 \mid y = -1] + \Pr[x + \theta > 0 \mid y = +1]\Big) \\
&= \frac{1}{2}\Big(\Pr[\mathcal{N}(0, 1 + \sigma^2) < 0] + \Pr[\mathcal{N}(2, 1 + \sigma^2) > 0]\Big) \\
&= \frac{1}{2}\Big(\Phi'(0) + \Phi'(2)\Big)
\end{aligned}
\tag{20}
$$

where $\Phi'(x) := \Phi(x/\sqrt{1 + \sigma^2})$ is the cumulative distribution function of $\mathcal{N}(0, 1 + \sigma^2)$.

We then compute its probability of making robustly correct predictions (i.e., dotted area)

$$\Pr[F_\theta(x + \delta) = y \wedge F_\theta(x) = y]$$

$$= \frac{1}{2}\Big(\Pr[F_\theta(x + \delta) = y \wedge F_\theta(x) = y \,|\, y = -1] + \Pr[F_\theta(x + \delta) = y \wedge F_\theta(x) = y \,|\, y = +1]\Big)$$

$$= \frac{1}{2}\Big(\Pr[x + \theta + \epsilon < 0 \wedge x + \theta < 0 \,|\, y = -1] + \Pr[x + \theta - \epsilon > 0 \wedge x + \theta > 0 \,|\, y = +1]\Big)$$

(where we use $x + \theta - \epsilon$ when $y = +1$ because the correctly classified sample must lie on the right)

$$= \frac{1}{2}\Big(\Pr[x + \theta < -\epsilon \,|\, y = -1] + \Pr[x + \theta > \epsilon \,|\, y = +1]\Big)$$

$$= \frac{1}{2}\Big(\Pr[\mathcal{N}(0, 1 + \sigma^2) < -\epsilon] + \Pr[\mathcal{N}(2, 1 + \sigma^2) > \epsilon]\Big)$$

$$= \frac{1}{2}\Big(\Phi'(-\epsilon) + \Phi'(2 - \epsilon)\Big),$$

(21)

where $\Phi'(x) := \Phi(x/\sqrt{1 + \sigma^2})$ is the cumulative distribution function of $\mathcal{N}(0, 1 + \sigma^2)$.

It shows that this classifier has robust accuracy (i.e., dotted area over shadowed area)

$$\Pr[F_\theta(x + \delta) = y \,|\, F_\theta(x) = y] = \frac{\Pr[F_\theta(x + \delta) = y \wedge F_\theta(x) = y]}{F_\theta(x) = y} = \frac{\Phi'(-\epsilon) + \Phi'(2 - \epsilon)}{\Phi'(0) + \Phi'(2)},$$

(22)

where $\Phi'(x) := \Phi(x/\sqrt{1 + \sigma^2})$ is the cumulative distribution function of $\mathcal{N}(0, 1 + \sigma^2)$. This verifies the computation in Equation (5).

Here, we can make a quick observation under Assumption 4, where we assume $\sigma = 1$ and $\epsilon = 1$ for simplicity. It shows that the stochastic pre-processing defense in our setting explicitly reduces invariance and utility to gain robustness. The general case is proven in Theorem 1.

**Observation 1.** *The defended classifier $F_\theta(x) = \text{sgn}(x + \theta)$ has higher robust accuracy (67.7% vs. 59.4%) yet lower benign accuracy (71.1% vs. 84.1%) than the undefended classifier $F(x) = \text{sgn}(x)$.*

### C.1.3 Defended Classification (Trained Invariance)

One critical step of stochastic pre-processing defenses is to preserve the defended model's utility by minimizing the risk over processed data $t_\theta(x)$, which leads to a new defended classifier $F_\theta^+(x) = \text{sgn}(x + \theta - 1)$ that is optimal on transformed data, as illustrated in Figure 6c. It has benign accuracy

$$\Pr[F_\theta^+(x) = y] = \frac{1}{2}\Big(\Pr[F_\theta^+(x) = y \,|\, y = -1] + \Pr[F_\theta^+(x) = y \,|\, y = +1]\Big)$$

$$= \frac{1}{2}\Big(\Pr[x + \theta - 1 < 0 \,|\, y = -1] + \Pr[x + \theta - 1 > 0 \,|\, y = +1]\Big)$$

$$= \frac{1}{2}\Big(\Pr[\mathcal{N}(0, 1 + \sigma^2) < 1] + \Pr[\mathcal{N}(2, 1 + \sigma^2) > 1]\Big)$$

$$= \frac{1}{2}\Big(\Phi'(1) + 1 - \Phi'(-1)\Big)$$

$$= \Phi'(1),$$

(23)

where $\Phi'(x) := \Phi(x/\sqrt{1 + \sigma^2})$ is the cumulative distribution function of $\mathcal{N}(0, 1 + \sigma^2)$.

We then compute its probability of making robustly correct predictions (i.e., dotted area)

$$\Pr[F_\theta^+(x+\delta) = y \wedge F_\theta^+(x) = y]$$

$$= \frac{1}{2}\Big(\Pr[F_\theta^+(x+\delta) = y \wedge F_\theta^+(x) = y \mid y = -1] + \Pr[F_\theta^+(x+\delta) = y \wedge F_\theta^+(x) = y \mid y = +1]\Big)$$

$$= \frac{1}{2}\Big(\Pr[x+\theta-1+\epsilon < 0 \wedge x+\theta-1 < 0 \mid y = -1] + \Pr[x+\theta-1-\epsilon > 0 \wedge x+\theta-1 > 0 \mid y = +1]\Big)$$

(where we use $x+\theta-1-\epsilon$ when $y = +1$ because the correctly classified sample must lie on the right)

$$= \frac{1}{2}\Big(\Pr[x+\theta < 1-\epsilon \mid y = -1] + \Pr[x+\theta > 1+\epsilon \mid y = +1]\Big)$$

$$= \frac{1}{2}\Big(\Pr[\mathcal{N}(0, 1+\sigma^2) < 1-\epsilon] + \Pr[\mathcal{N}(2, 1+\sigma^2) > 1+\epsilon]\Big)$$

$$= \frac{1}{2}\Big(\Phi'(1-\epsilon) + 1 - \Phi'(\epsilon-1)\Big)$$

$$= \Phi'(1-\epsilon),$$

(24)

where $\Phi'(x) := \Phi(x/\sqrt{1+\sigma^2})$ is the cumulative distribution function of $\mathcal{N}(0, 1+\sigma^2)$.

It shows that this classifier has robust accuracy (i.e., dotted area over shadowed area)

$$\Pr[F_\theta^+(x+\delta) = y \mid F_\theta^+(x) = y] = \frac{\Pr[F_\theta^+(x+\delta) = y \wedge F_\theta^+(x) = y]}{F_\theta^+(x) = y} = \frac{\Phi'(1-\epsilon)}{\Phi'(1)}, \quad (25)$$

where $\Phi'(x) := \Phi(x/\sqrt{1+\sigma^2})$ is the cumulative distribution function of $\mathcal{N}(0, 1+\sigma^2)$. This verifies the computation in Equation (6).

Here, we can make the following two observations under Assumption 4, where we assume $\sigma = 1$ and $\epsilon = 1$ for simplicity. They show that the defense has to increase the invariance, which was previously reduced to gain robustness, to recover utility. The general case is proven in Theorem 1.

**Observation 2.** *The defended classifier with trained invariance $F_\theta^+(x) = \mathrm{sgn}(x+\theta-1)$ is less robust (65.8% vs. 70.4%) than the defended classifier $F_\theta(x) = \mathrm{sgn}(x+\theta)$ without trained invariance.*

**Observation 3.** *The defended classifier with trained invariance $F_\theta^+(x) = \mathrm{sgn}(x+\theta-1)$ is more robust (65.8% vs. 59.4%) than the original undefended classifier $F(x) = \mathrm{sgn}(x)$ at the cost of utility (76.0% vs. 84.1%).*

### C.1.4 Defended Classification (Perfect Invariance)

Furthermore, these defenses usually leverage majority vote to obtain stable predictions, which finally produces a perfectly invariant defended classifier:

$$F_\theta^*(x) = \underset{y \in \{-1, +1\}}{\arg\max} \sum_{i=1}^{n} \mathbb{1}\big\{F_{\theta_i}^+(x) = y\big\}$$

$$= \mathrm{sgn}\left(\frac{1}{n}\sum_{i=1}^{n} F_{\theta_i}^+(x)\right)$$

$$= \mathrm{sgn}\left(\frac{1}{n}\sum_{i=1}^{n} \mathrm{sgn}(x+\theta_i-1)\right)$$

$$\rightarrow \mathrm{sgn}\left(\underset{\theta \sim \mathcal{N}(1, \sigma^2)}{\mathbb{E}}\Big[\mathrm{sgn}(x+\theta-1) \,\Big|\, x\Big]\right)$$

$$= \mathrm{sgn}\left(\underset{z|x \sim \mathcal{N}(x, \sigma^2)}{\mathbb{E}}\Big[\mathrm{sgn}(z) \,\Big|\, x\Big]\right)$$

$$= \mathrm{sgn}\left(\Pr\Big[\mathcal{N}(x, \sigma^2) > 0 \,\Big|\, x\Big] - \Pr\Big[\mathcal{N}(x, \sigma^2) < 0 \,\Big|\, x\Big]\right)$$

$$= \mathrm{sgn}(x), \quad (26)$$

where the last equality holds because $\mathcal{N}(x, \sigma^2)$ has more probability on the positive side if and only if $x > 0$ and has more probability on the negative side if and only if $x < 0$. As we can observe,

the defended classifier with trained invariance and majority vote reduces to the original undefended classifier $F(x) = \text{sgn}(x)$, which verifies Equation (7).

## C.2 Theorem: Trade-off between Robustness and Invariance

In this section, we extend the above coupling between robustness and invariance to a general trade-off, where we can control the invariance through shifting decision boundary and employing majority vote.

Recall that $x|y \sim \mathcal{N}(y, 1)$ and $\theta \sim \mathcal{N}(1, \sigma^2)$, we denote their density functions by

$$\varphi_x = \begin{cases} \varphi(x+1), & y = -1 \\ \varphi(x-1), & y = +1 \end{cases}, \Phi_x = \begin{cases} \Phi(x+1), & y = -1 \\ \Phi(x-1), & y = +1 \end{cases}, \varphi_\theta = \varphi\left(\frac{\theta - 1}{\sigma}\right), \Phi_\theta = \Phi\left(\frac{\theta - 1}{\sigma}\right),$$
(27)

where $\varphi$ and $\Phi$ are the probability and cumulative density functions of $\mathcal{N}(0,1)$, respectively.

**Rate of Invariance.** To facilitate our analysis, given the theoretical setting and assumptions specified in Appendix C.1, we define the *rate of invariance* for a defended classifier $F_\theta(x)$ as

$$R(k) := \Pr[F_\theta(x) = F(x)],$$
(28)

where $F_\theta(x) = \text{sgn}(x + \theta - k)$, and $F(x) = \text{sgn}(x)$ is the undefended classifier.

We formalize the trade-off between robustness and invariance in the following theorem proven in Appendix C.3.4. It shows that stochastic pre-processing defenses provide robustness by intentionally reducing the model's invariance to added randomized transformations.

**Theorem 1** (Trade-off between Robustness and Invariance). *Given the above theoretical setting and assumptions, when the defended classifier $F_\theta(x)$ achieves higher invariance $R(k)$ under the defense's randomization space to preserve utility, the adversarial robustness provided by the defense strictly decreases.*

We prove this theorem by characterizing the (strictly opposite) monotonic behavior of invariance and robustness as the defended classifier shifts its decision boundary towards the optimal decision boundary $k = 1$ on transformed data (see Appendix C.1.3) and applies majority vote at the end. We formalize such characterizations in the following lemmas and corollaries.

First, we show in Lemma 1 that the defended classifier's rate of invariance strictly increases as the decision boundary shifts towards the optima; applying majority vote further yields perfect invariance, as we show in Corollary 1. We prove them in Appendix C.3.1.

**Lemma 1** (Strictly Increasing Invariance). *The defended classifier's invariance $R(k)$ strictly increases as the decision boundary approaches $k = 1$ without applying majority vote.*

**Corollary 1** (Perfect Invariance by Majority Vote). *When the defended classifier maximizes trained invariance at $k = 1$, employing majority vote further improves the rate of invariance $R(k)$ to one.*

Second, we show in Lemma 2 that the defended classifier's robust accuracy strictly decreases as the decision boundary shifts the optima. When the trained invariance is approximated, we show in Corollary 2 that applying majority vote strictly decreases the robust accuracy further. We prove them in Appendix C.3.2.

**Lemma 2** (Strictly Decreasing Robustness). *The defended classifier's robust accuracy strictly decreases as the decision boundary approaches $k = 1$ without applying majority vote.*

**Corollary 2** (Strictly Decreasing Robustness by Majority Vote). *When the defended classifier approximates the trained invariance by shifting its decision boundary to $k \in [0, 2]$, applying majority vote strictly decreases its robust accuracy.*

Finally, we show in Lemma 3 that the defended classifier indeed preserves its utility by shifting the decision boundary towards the optima, and applying majority vote recovers the full utility as we show in Corollary 3. We prove them in Appendix C.3.3.

**Lemma 3** (Strictly Increasing Accuracy). *The defended classifier's benign accuracy strictly increases as the decision boundary approaches $k = 1$ without applying majority vote.*

**Corollary 3** (Strictly Increasing Accuracy by Majority Vote). *When the defended classifier approximates the trained invariance by shifting its decision boundary to $k \in [0, 2]$, applying majority vote strictly increases its accuracy.*

### C.3 Proofs

We provide complete proofs for the theorems, lemmas, and corollaries that we present above.

#### C.3.1 Strictly Increasing Invariance

**Lemma 1** (Strictly Increasing Invariance). *The defended classifier's invariance $R(k)$ strictly increases as the decision boundary approaches $k = 1$ without applying majority vote.*

*Proof.* We directly compute the rate of invariance $R(k)$ as

$$
\begin{aligned}
R(k) &= \Pr[F_\theta(x) = F(x)] \\
&= \Pr[\operatorname{sgn}(x + \theta - k) = \operatorname{sgn}(x)] \\
&= \Pr[\operatorname{sgn}(x + \theta - k) = \operatorname{sgn}(x) \wedge x < 0] + \Pr[\operatorname{sgn}(x + \theta - k) = \operatorname{sgn}(x) \wedge x > 0] \\
&= \Pr[\theta < k - x \wedge x < 0] + \Pr[\theta > k - x \wedge x > 0] \\
&= \int_{-\infty}^0 \int_{-\infty}^{k-x} \varphi_x(x) \cdot \varphi_\theta(\theta) \, \mathrm{d}\theta \, \mathrm{d}x + \int_0^\infty \int_{k-x}^\infty \varphi_x(x) \cdot \varphi_\theta(\theta) \, \mathrm{d}\theta \, \mathrm{d}x \\
&= \int_{-\infty}^0 \varphi_x(x) \cdot \Phi_\theta(k - x) \, \mathrm{d}x - \int_0^\infty \varphi_x(x) \cdot \Phi_\theta(k - x) \, \mathrm{d}x + \int_0^\infty \varphi_x(x) \, \mathrm{d}x,
\end{aligned}
\tag{29}
$$

whose gradient with respect to $k$ is

$$
\begin{aligned}
\frac{\partial}{\partial k} R(k) &= \int_{-\infty}^0 \varphi_x(x) \cdot \varphi_\theta(k - x) \, \mathrm{d}x - \int_0^\infty \varphi_x(x) \cdot \varphi_\theta(k - x) \, \mathrm{d}x \\
&= \frac{1}{2} \left( \int_{-\infty}^0 \varphi(x + 1) \cdot \varphi_\theta(k - x) \, \mathrm{d}x - \int_0^\infty \varphi(x + 1) \cdot \varphi_\theta(k - x) \, \mathrm{d}x \right) \\
&\quad + \frac{1}{2} \left( \int_{-\infty}^0 \varphi(x - 1) \cdot \varphi_\theta(k - x) \, \mathrm{d}x - \int_0^\infty \varphi(x - 1) \cdot \varphi_\theta(k - x) \, \mathrm{d}x \right).
\end{aligned}
\tag{30}
$$

From calculus and the error function $\operatorname{erf}$ we have

$$
\begin{aligned}
p_1(x) &:= \int \varphi(x + 1) \cdot \varphi_\theta(k - x) \, \mathrm{d}x = \frac{1}{4\sqrt{\pi}} \exp\left( -\frac{k^2}{4} \right) \cdot \operatorname{erf}\left( 1 - \frac{k}{2} + x \right) \\
p_2(x) &:= \int \varphi(x - 1) \cdot \varphi_\theta(k - x) \, \mathrm{d}x = \frac{-1}{4\sqrt{\pi}} \exp\left( -\frac{(k-2)^2}{4} \right) \cdot \operatorname{erf}\left( \frac{k}{2} - x \right),
\end{aligned}
\tag{31}
$$

where we have assumed $\theta \sim \mathcal{N}(1, 1)$ to simplify the analysis by Assumption 4.

It shows that the gradient in Equation (30) is

$$
\begin{aligned}
\frac{\partial}{\partial k} R(k) &= \frac{1}{2} \Big( p_1(0) - p_1(-\infty) - p_1(\infty) + p_1(0) \Big) + \frac{1}{2} \Big( p_2(0) - p_2(-\infty) - p_2(\infty) + p_2(0) \Big) \\
&= p_1(0) + p_2(0) \\
&\propto \exp\left( -\frac{k^2}{4} \right) \cdot \operatorname{erf}\left( 1 - \frac{k}{2} \right) - \exp\left( -\frac{(k-2)^2}{4} \right) \cdot \operatorname{erf}\left( \frac{k}{2} \right).
\end{aligned}
\tag{32}
$$

Notice that $G(k) := \frac{\partial}{\partial k} R(k)$ is a symmetric function with respect to the point $(1, 0)$:

$$
G(1 + z) + G(1 - z) = 0, \quad \forall z \in \mathbb{R},
\tag{33}
$$

which shows that $G(k)$ attains zero at $k = 1$.

Since both $\exp$ and $\operatorname{erf}$ are strictly increasing functions, for $k < 1$, we have

$$
\begin{aligned}
-\frac{k^2}{4} > -\frac{(k-2)^2}{4} &\implies \exp\left( -\frac{k^2}{4} \right) > \exp\left( -\frac{(k-2)^2}{4} \right), \\
1 - \frac{k}{2} > \frac{k}{2} &\implies \operatorname{erf}\left( 1 - \frac{k}{2} \right) > \operatorname{erf}\left( \frac{k}{2} \right),
\end{aligned}
\tag{34}
$$

which shows that $G(k) > 0$ when $k < 1$, and $G(k) < 0$ when $k > 1$ by symmetry.

Therefore, the rate of invariance $R(k)$ strictly increases for $k < 1$ and strictly decreases for $k > 1$. $\square$

**Corollary 1** (Perfect Invariance by Majority Vote). *When the defended classifier maximizes trained invariance at $k = 1$, employing majority vote further improves the rate of invariance $R(k)$ to one.*

*Proof.* We showed in Appendix C.1.4 that the defended classifier $F_\theta(x) = \mathrm{sgn}(x + \theta - 1)$ converges to the optimal classifier $F_\theta(x) = \mathrm{sgn}(x)$ if given a sufficiently large number of votes.

In such a case, it is straightforward to show that the rate of invariance converges to one:

$$R(k = 1) = \Pr[F_\theta(x) = F(x)] \to \Pr[\mathrm{sgn}(x) = \mathrm{sgn}(x)] = 1. \tag{35}$$

$\square$

### C.3.2 Strictly Decreasing Robustness

**Lemma 2** (Strictly Decreasing Robustness). *The defended classifier's robust accuracy strictly decreases as the decision boundary approaches $k = 1$ without applying majority vote.*

*Proof.* We directly compute the robust accuracy of the defended classifier $F_\theta(x) = \mathrm{sgn}(x + \theta - k)$ and characterize its monotonic behavior. Recall that $x + \theta \sim \mathcal{N}(y + 1, 2)$.

We first compute the defended classifier's benign accuracy:

$$
\begin{aligned}
&\Pr[F_\theta(x) = y] \\
&= \Pr[\mathrm{sgn}(x + \theta - k) = y] \\
&= \Pr[x + \theta < k \mid y = -1] \cdot \Pr[y = -1] + \Pr[x + \theta > k \mid y = +1] \cdot \Pr[y = +1] \\
&= \Pr[\mathcal{N}(0, 2) < k] \cdot \Pr[y = -1] + \Pr[\mathcal{N}(2, 2) > k] \cdot \Pr[y = +1] \\
&= \frac{1}{2}\Big(\Phi'(k) + \Phi'(2 - k)\Big),
\end{aligned}
\tag{36}
$$

where $\Phi'$ denotes the cumulative density function of $\mathcal{N}(0, 2)$.

We then compute the probability of robustly correct predictions, where we use $x + \delta$ to denote the adversarial example that actually can take any value from $[x - \epsilon, x + \epsilon]$ to change the prediction:

$$
\begin{aligned}
&\Pr[F_\theta(x + \delta) = y \wedge F_\theta(x) = y] \\
&= \frac{1}{2}\Big(\Pr[F_\theta(x + \delta) = y \wedge F_\theta(x) = y \mid y = -1] + \Pr[F_\theta(x + \delta) = y \wedge F_\theta(x) = y \mid y = +1]\Big) \\
&= \frac{1}{2}\Big(\Pr[x + \theta - k + \epsilon < 0 \wedge x + \theta - k < 0 \mid y = -1] + \Pr[x + \theta - k - \epsilon > 0 \wedge x + \theta - k > 0 \mid y = +1]\Big) \\
&\quad \text{(where we use } -\epsilon \text{ when } y = +1 \text{ because the correctly classified sample must lie on the right)} \\
&= \frac{1}{2}\Big(\Pr[x + \theta < k - \epsilon \wedge x + \theta < k \mid y = -1] + \Pr[x + \theta > k + \epsilon \wedge x + \theta > k \mid y = +1]\Big) \\
&= \frac{1}{2}\Big(\Pr[x + \theta < k - \epsilon \mid y = -1] + \Pr[x + \theta > k + \epsilon \mid y = +1]\Big) \\
&= \frac{1}{2}\Big(\Pr[\mathcal{N}(0, 2) < k - \epsilon] + \Pr[\mathcal{N}(2, 2) > k + \epsilon]\Big) \\
&= \frac{1}{2}\Big(\Phi'(k - \epsilon) + \Phi'(2 - k - \epsilon)\Big),
\end{aligned}
\tag{37}
$$

where $\Phi'$ denotes the cumulative density function of $\mathcal{N}(0, 2)$.

Now we can compute the robust accuracy at decision boundary $k$ as

$$\Pr[F_\theta(x + \delta) = y \mid F_\theta(x) = y] = \frac{\Pr[F_\theta(x + \delta) = y \wedge F_\theta(x) = y]}{\Pr[F_\theta(x) = y]} = \frac{\Phi'(k - \epsilon) + \Phi'(2 - k - \epsilon)}{\Phi'(k) + \Phi'(2 - k)}. \tag{38}$$

While the argument holds for any fixed $\epsilon$, we will show a simple example and assume a reasonably strong adversary with $\epsilon = 1$ (Assumption 4), which initializes the robust accuracy to:

$$\texttt{Rob}(k) \coloneqq \frac{\Phi'(k-1) + \Phi'(1-k)}{\Phi'(k) + \Phi'(2-k)}, \tag{39}$$

where $\Phi'$ denotes the cumulative density function of $\mathcal{N}(0, 2)$. Its gradient with respect to $k$ is

$$G(k) \coloneqq \frac{\partial}{\partial k}\texttt{Rob}(k) = \frac{2\exp\left(-1 - \frac{k^2}{4}\right)(e^k - e)}{\sqrt{\pi}\left(2 - \text{erf}\left(\frac{k-2}{2}\right) - \text{erf}\left(-\frac{k}{2}\right)\right)^2} \propto e^k - e, \tag{40}$$

which shows that $G(k) < 0$ when $k < 1$, $G(k) > 0$ when $k > 1$, and $G(k) = 0$ when $k = 1$.

Therefore, the robust accuracy $\texttt{Rob}(k)$ strictly decreases as $k$ approaches $k = 1$ from either side. $\square$

**Corollary 2** (Strictly Decreasing Robustness by Majority Vote). *When the defended classifier approximates the trained invariance by shifting its decision boundary to $k \in [0, 2]$, applying majority vote strictly decreases its robust accuracy.*

*Proof.* For this proof, we assume the applied defense adopts $\theta \sim \mathcal{N}(1, \sigma^2)$, which reformulates the robust accuracy in Equation (39) as

$$\texttt{Rob}(k) \coloneqq \frac{\Phi'(k-1) + \Phi'(1-k)}{\Phi'(k) + \Phi'(2-k)} = 2\left(2 + \text{erf}\left(\frac{2-k}{\sqrt{2}\sqrt{1+\sigma^2}}\right) + \text{erf}\left(\frac{k}{\sqrt{2}\sqrt{1+\sigma^2}}\right)\right)^{-1}, \tag{41}$$

where $\Phi'$ is the cumulative density function of $\mathcal{N}(0, 1 + \sigma^2)$.

For $k \in [0, 2]$, where the argument for erf is non-negative, decreasing $\sigma$ will also decrease the robust accuracy (the erf function is strictly increasing). Given that majority vote effectively reduces the noise's variance, having a larger number of votes will strictly decrease the robust accuracy. $\square$

### C.3.3 Strictly Increasing Accuracy

**Lemma 3** (Strictly Increasing Accuracy). *The defended classifier's benign accuracy strictly increases as the decision boundary approaches $k = 1$ without applying majority vote.*

*Proof.* In Equation (36), we showed that the defended classifier $F_\theta(x) = \text{sgn}(x + \theta - k)$ has benign accuracy

$$\texttt{Acc}(k) \coloneqq \Pr[F_\theta(x) = y] = \frac{1}{2}\left(\Phi'(k) + \Phi'(2-k)\right), \tag{42}$$

where $\Phi'$ denotes the cumulative density function of $\mathcal{N}(0, 2)$.

Its gradient with respect to $k$ is

$$G(k) \coloneqq \frac{\partial}{\partial k}\texttt{Acc}(k) = \frac{\exp\left(-1 - \frac{k^2}{4}\right)(e - e^k)}{2\sqrt{\pi}} \propto e - e^k, \tag{43}$$

which shows that $G(k) > 0$ when $k < 1$, $G(k) < 0$ when $k > 1$, and $G(k) = 0$ when $k = 1$.

Therefore, the benign accuracy $\texttt{Acc}(k)$ strictly increases as $k$ approaches $k = 1$ from either side. $\square$

**Corollary 3** (Strictly Increasing Accuracy by Majority Vote). *When the defended classifier approximates the trained invariance by shifting its decision boundary to $k \in [0, 2]$, applying majority vote strictly increases its accuracy.*

*Proof.* For this proof, we assume the applied defense adopts $\theta \sim \mathcal{N}(1, \sigma^2)$, which reformulates the benign accuracy in Equation (42) as

$$\texttt{Acc}(k) \coloneqq \frac{1}{2}\left(\Phi'(k) + \Phi'(2-k)\right) = 2 + \text{erf}\left(\frac{2-k}{\sqrt{2}\sqrt{1+\sigma^2}}\right) + \text{erf}\left(\frac{k}{\sqrt{2}\sqrt{1+\sigma^2}}\right), \tag{44}$$

where $\Phi'$ denotes the cumulative density function of $\mathcal{N}(0, 1 + \sigma^2)$.

For $k \in [0, 2]$, where the argument for erf is non-negative, decreasing $\sigma$ will also decrease the robust accuracy (the erf function is strictly increasing). Given that majority vote effectively reduces the noise's variance, having a larger number of votes will strictly increase the robust accuracy.

As a special case, when $k = 1$ and $\sigma \to 0$, we have

$$\text{Acc}(k) = \frac{1}{2}\Big(\Phi'(k) + \Phi'(2 - k)\Big) = \frac{1}{2}\left(\Phi\left(\frac{k}{\sqrt{1 + \sigma^2}}\right) + \Phi\left(\frac{2 - k}{\sqrt{1 + \sigma^2}}\right)\right) \to \Phi(1), \quad (45)$$

which recovers the full utility of the undefended classifier in Equation (17). $\qquad\square$

### C.3.4 Trade-off between Robustness and Invariance

**Theorem 1** (Trade-off between Robustness and Invariance). *Given the above theoretical setting and assumptions, when the defended classifier $F_\theta(x)$ achieves higher invariance $R(k)$ under the defense's randomization space to preserve utility, the adversarial robustness provided by the defense strictly decreases.*

*Proof.* The proof follows by directly combining the lemmas and corollaries proven above.

By Lemma 3 and Corollary 3, when the defended classifier $F_\theta(x) = x + \theta - k$ shifts its decision boundary towards $k = 1$, its benign accuracy strictly increases and is maximized at $k = 1$ with the application of majority vote. This verifies that the defended classifier in our setting indeed preserves utility by shifting the decision boundary towards $k = 1$.

By Lemma 1 and Corollary 1, when the defended classifier shifts the decision boundary towards $k = 1$ to preserve utility, its rate of invariance strictly increases and is maximized at $k = 1$ with the application of majority vote. This verifies that the defended classifier in our setting strictly controls its invariance by shifting the decision boundary.

By Lemma 2 and Corollary 2, when the defended classifier shifts the decision boundary towards $k = 1$ to acquire more invariance, the adversarial robustness strictly decreases and is minimized at $k = 1$ with the application of majority vote.

The above arguments show that the defended classifier strictly improves its invariance by approaching $k = 1$, yet the adversarial robustness strictly decreases during this process. When perfect invariance is achieved, the utility and robustness go back to those of the undefended classifier, nullifying the initially applied stochastic pre-processing defense. $\qquad\square$

## D   Experiment Setup: Main Evaluation

In this section, we provide more details of our main evaluation.

### D.1   Datasets

We conduct all experiments on the public ImageNet [30] and ImageNette [9] datasets.

For ImageNet, our test data consists of 1,000 images randomly sampled from the validation set. These images are only sampled once and are fixed for all experiments. We did not train models on the ImageNet training data in our experiments.

ImageNette is a ten-class subset of ImageNet. Its original training set and validation set have 9,469 and 3,925 images, respectively. We randomly split its original training set into our 90% and 10% training and validation data, and adopt 1,000 images randomly sampled from its original validation set as our test data. The data split and test images are only sampled once and fixed for all experiments. We use the high-resolution version of ImageNette, where all images are larger than $320 \times 320$.

Because some of our experiments require fine-tuning models on processed training data, we switch to ImageNette to reduce the training cost. We evaluate on ImageNet only when (1) model fine-tuning is not needed, or (2) model fine-tuning is needed but a pre-trained model is publicly available.

## D.2 Models

We adopt various ResNet [13] models mainly depending on the examined defense. All models make the prediction with majority vote over $n = 500$ samples if a stochastic defense is applied.

For defenses with low randomness, which require no model fine-tuning, we evaluate them on ImageNet with a ResNet-50 model pre-trained by TorchVision[7], which attains 76.13% Top-1 accuracy and 92.86% Top-5 accuracy on ImageNet.

For defenses with higher randomness, which require model fine-tuning, we evaluate them on ImageNette with our own ResNet-34 models, detailed as follows.

To first obtain a baseline model for ImageNette, we adopt a ResNet-34 model pre-trained by TorchVision, which attains 73.31% Top-1 accuracy and 91.42% Top-5 accuracy on ImageNet. We fine-tune this model on ImageNette's training set with gradient descent for 70 epochs using the AdamW [21] optimizer and the Cosine Annealing [20] learning rate scheduler, where we use batch size 256, initial learning rate 0.001, and weight decay 0.01. We choose the model that performs best on the validation set, which attains 96.9% Top-1 accuracy on the test set.

We then fine-tune the above baseline ResNet-34 model on training data pre-processed by the defense we examine in each experiment. We adopt the same training configs as those used to train the baseline model but reduce the number of epochs to 30.

As a special case, when we evaluate randomized smoothing in Section 6.2, which requires model fine-tuning with data perturbed by Gaussian noise, we adopt the ResNet-50 models pre-trained on such perturbed ImageNet from Cohen et al. [6]. These models attain 67% and 57% Top-1 accuracy when the input is perturbed with Gaussian noise of standard deviation 0.25 and 0.50, respectively.

## D.3 Defenses

Our main evaluation focuses on two stochastic defenses, detailed as follows.

**BaRT [28].** The original BaRT defense considers a randomization space of 25 diverse input transformations, and the parameters of each transformation are further randomized. At each inference, it randomly samples $\kappa$ randomized transformations, composites them together in a random order, and applies the composited transformation to the input image.

Since our evaluation only aims to examine the limitations of BaRT but not to break it, it suffices to analyze a *subset* of transformations. Specifically, we consider a randomization space of $\kappa \leq 6$ input transformations and composite all $\kappa$ transformations in a random order to pre-process the input image before feeding it to the classifier. We outline the chosen randomized transformations below and refer to Raff et al. [28] for more details. Our implementation is available in the code.

- *Noise Injection.* This transformation perturbs the input image with noise of distributions and parameters chosen uniformly at random. The set of candidate noise distributions includes Gaussian, Poisson, Salt, Pepper, Salt and Pepper, and Speckle.

- *Gaussian Blur.* This transformation blurs the input image using a Gaussian filter with the kernel size randomly chosen from $[2, 14]$ and the standard deviation randomly chosen from $[0.1, 3.1]$.

- *Median Blur.* This transformation blurs the input image using a median filter with the kernel size randomly chosen from $[2, 14]$.

- *Swirl Transformation.* This transformation applies the swirl transformation[8], a non-linear image deformation that creates a whirlpool effect. Its strength, radius, and location are chosen uniformly at random from $[0.1, 2.0]$, $[10, 200]$, and $[1, 200]$, respectively. We adopt BPDA [2] with the identity function to handle the non-differentiable problem.

- *Quantization.* This transformation quantizes the input image's pixel values within $[0, 1]$ to a limited number of bins, where the number of bins is chosen uniformly at random from $[8, 200]$. For example, if the number of bins is set to 4, all pixels will be quantized to $\{0.00, 0.25, 0.50, 0.75, 1.00\}$.

---

[7] https://pytorch.org/vision/stable/models.html
[8] https://scikit-image.org/docs/stable/auto_examples/transform/plot_swirl.html

- *FFT Perturbation.* This transformation perturbs the 2D FFT of each channel of the input image. For each channel in the frequency domain, it randomly zeros out a fraction of coefficients. The fraction is chosen uniformly at random from $[0.00, 0.95]$.

In our setting, we form the randomization space by compositing the first $\kappa$ transformations in random order. While increasing the space of transformations typically leads to a more effective defense, randomly compositing transformations may not always lead to stronger defenses. For example, the quantization may decrease the effectiveness of other transformations. However, this drawback *does not* affect our evaluation, as our main objective is to compare the defense's performance before and after the defended model achieves higher invariance. Rigorous comparisons between the defense's performance before and after increasing the randomness are largely orthogonal to our work.

**Randomized Smoothing [6].** Randomized smoothing adds Gaussian noise to the input image and makes predictions with majority vote over a large number of samples. This defense was initially proposed for certifiable adversarial robustness. In our evaluation, we adopt this defense to examine (1) how randomness affects the effectiveness of applying EOT and (2) how invariance affects the robustness provided by the defense. Specifically, we control the level of randomness by varying the added Gaussian noise's standard deviation $\sigma$. For evaluation on ImageNet, we choose $\sigma \in \{0.25, 0.50\}$ as models pre-trained on data perturbed by such noise are available from Cohen et al. [6]. For evaluation on ImageNette, we are able to scale the evaluation for $\sigma$ from 0.10 to 0.50 with a step size of 0.05. We ignore the *abstain* output in the original defense, as we do not study the certification.

## D.4 Attacks

We evaluate defenses with standard PGD combined with EOT and focus on the $\ell_\infty$-bounded adversary with a perturbation budget $\epsilon = 8/255$ in both untargeted and targeted settings. We do not introduce any techniques other than EOT to explicitly handle the randomness, such as random restarts [7] and momentum-based optimizers [34]. We also utilize AutoPGD [7] to avoid selecting the best step size when it is computationally expensive to repeat some experiments. More importantly, we only conduct adaptive evaluations, where the defense is always included in the attack loop with non-differentiable components approximated by the identity function [2]. For targeted attacks, we choose the last class of each dataset as the target label: ImageNet (999) and ImageNette (9).

We adopt various attack settings depending on the experiment, as detailed below.

In Section 6.2, we aim to evaluate the benefits of applying EOT under different settings. For this end, we apply the standard PGD attack of $k \in \{10, 20, 50, 100, 200, 500, 1000\}$ steps and combine them with EOT of $m \in \{1, 5, 10, 20\}$ samples. For each combination, we further test several step sizes chosen from $\alpha \in \{0.5/255, 1/255, 2/255, 4/255\}$ and report their best performance.

In Section 6.3, we aim to evaluate the trade-off between the defense's robustness and the model's invariance to the added randomness. For this end, we evaluate the defense's performance when it applies to models of different levels of invariance. Since it is computationally expensive to repeat the attack for multiple step sizes, we utilize AutoPGD [7] to tune the step size automatically. In this experiment, we evaluate all defenses with AutoPGD of 200 steps and disable all techniques designed to capture the randomness, including EOT.

We make this choice due to three considerations. First, disabling EOT effectively reduces the computational cost. Second, we already showed in Section 6.2 that PGD attacks can already assess the robustness of stochastic defenses without applying EOT. Last but not least, our objective is to evaluate the defense's performance when it applies to different models *under the same attack*. Under this setting, we can observe that the same attack (regardless of its strength) that hardly works for the defense (before fine-tuning & low invariance) now becomes more effective (after fine-tuning & high invariance). We can surely run each attack for more iterations and samples, but the current setting suffices to show that the defense provides robustness by explicitly reducing invariance.

**Computing Resources.** All experiments are conducted on two Linux workstations, each with 48 Intel Xeon CPUs and 8 GeForce RTX 2080 Ti GPUs. We only train ResNet-34 models on ImageNette without the distributed setting. Standard training (70 epochs) takes 35 minutes. Training with data processed by Gaussian noise (30 epochs) takes 15 minutes. Training with data processed by BaRT (30 epochs) takes 18 minutes for $\kappa \in \{1, 2\}$, 70 minutes for $\kappa \in \{3, 4, 5\}$, and 3 hours for $\kappa \in \{6\}$.

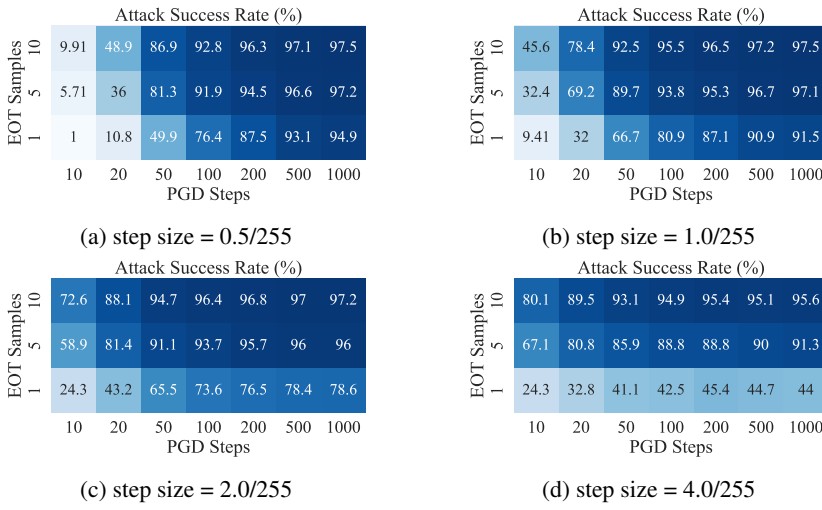

Figure 7: Evaluation of randomized smoothing on ImageNet (targeted attacks, $\sigma = 0.25$).

The training time for $\kappa \geq 3$ is higher due to CPU-bounded transformations; implementing such transformations using native PyTorch operations on GPUs should decrease the training cost.

# E  More Experiment Results

In this section, we provide more experiment results that strengthen discussions in the main paper.

## E.1  PGD Captures Randomness with Fine-grained Learning Rates

During our evaluation, we recognize that the effectiveness of PGD attacks, when combined with EOT, is sensitive to the choice of step size (i.e., learning rate). Here, we provide the full results of four different choices of step size when evaluating the randomized smoothing defense. The results with step size chosen from $\alpha \in \{0.5/255, 1.0/255, 2.0/255, 4.0/255\}$ are shown in Figure 7.

As we can observe, standard PGD without EOT achieves better performance when the step size is small, yet the application of EOT requires larger step sizes to perform better. We conjecture that this is because EOT reduces the variance of gradients so the attack algorithm can take a larger step, yet standalone PGD only gets noisy gradients and is only "confident" to take a small step.

However, this may not always prevent PGD from converging to a competitive solution. For example, we evaluate the discontinuous activation [40] defense with different attack settings, where the attack adds Gaussian noise around the input to estimate the correct gradient. The convergence curves of different settings are demonstrated in Figure 8.

When we examine the convergence in terms of PGD steps in Figure 8a, applying EOT obtains better gradients and quickly decreases the defended model's accuracy to zero. However, when we examine the convergence in terms of the total number of gradient queries in Figure 8b, we observe that (1) PGD without EOT given a smaller learning rate and (2) PGD with EOT given a larger learning rate have almost the same convergence behavior. This interesting observation suggests that standard PGD attacks may be sufficient in some cases if using a carefully fine-tuned learning rate. For example, Sitawarin et al. [34] showed that PGD attacks could be significantly improved by applying the AggMo [22] optimizer, which leverages multiple momentum terms.

## E.2  Inability to Remove Invariance that Does Not Hurt the Utility

Some recent works also suggest that one could gain robustness by removing invariance that does not hurt the utility [33]. However, this may not be the case for defenses with a larger randomization space. For example, we evaluate the performance of BaRT when it applies to models during the

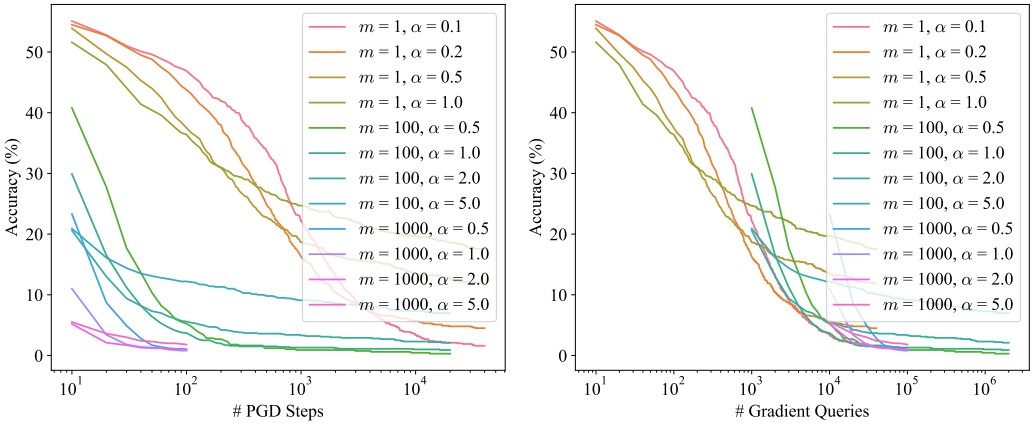

(a) Accuracy under Attack (view by PGD Steps)    (b) Accuracy under Attack (view by Gradient Queries)

Figure 8: Evaluation of the discontinuous activation [40] defense with EOT-$m$ and step size $\alpha$.

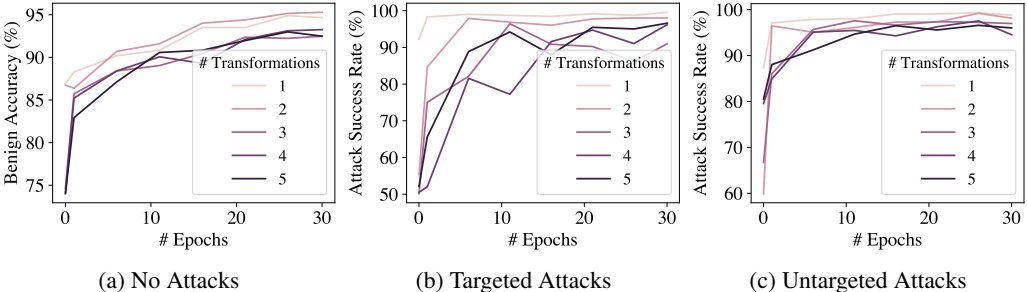

(a) No Attacks              (b) Targeted Attacks              (c) Untargeted Attacks

Figure 9: Performance of the BaRT defense on ImageNette and models during fine-tuning.

fine-tuning process (same experiment as in Section 6.3). As shown in Figure 9, the robustness has already dropped significantly before the model achieves invariance that preserves most of the utility.

### E.3    Visualization of Adversarial Perturbation

Figures 10 and 11 show the adversarial perturbation created by PGD attacks with and without EOT.

**Settings.** When given $C$ gradient queries in total, we run PGD for (1) $C$ steps without EOT and (2) $C/10$ steps with EOT of 10 samples. All attacks use $\ell_\infty$-norm budget $\epsilon = 8/255$ and step size $\alpha = 1/255$. The target model is a ResNet-50 defended by randomized smoothing with Gaussian noise $\sigma \in \{0.25, 0.50, 1.00\}$. We adopt pre-trained models from Cohen et al. [6].

**Visualization.** We randomly choose an image (id 5000) from the ImageNet validation set. For the benign image $x$ and its adversarial example $x'$, the perturbation is written as $\delta := x' - x \in [-\epsilon, \epsilon]$. We normalize it to $\delta' := \delta/(2\epsilon) + 0.5 \in [0, 1]$ and multiply it by 0.95 for better visualization.

**Compare PGD with and without EOT.** For models of the same noise level, applying EOT leads to slightly smoother (or less noisy) adversarial perturbation. This observation shows that EOT computes more stable gradients. Besides, the above effect becomes more significant when (1) the model has a higher level of randomness (i.e., large $\sigma$), or (2) the attack runs in the targeted mode. These are the scenarios where applying EOT benefits more, which correspond to our findings in Section 6.2.

**Compare PGD on models with different degrees of randomness.** If we compare the visualization across different models, we can observe that models with a higher level of randomness produce smoother adversarial perturbation (even without applying EOT). While this observation seems counterintuitive, we note that these models are all fine-tuned on noisy data. As a result, making the model invariant to randomness also smoothes out the gradient, which removes the expected robustness

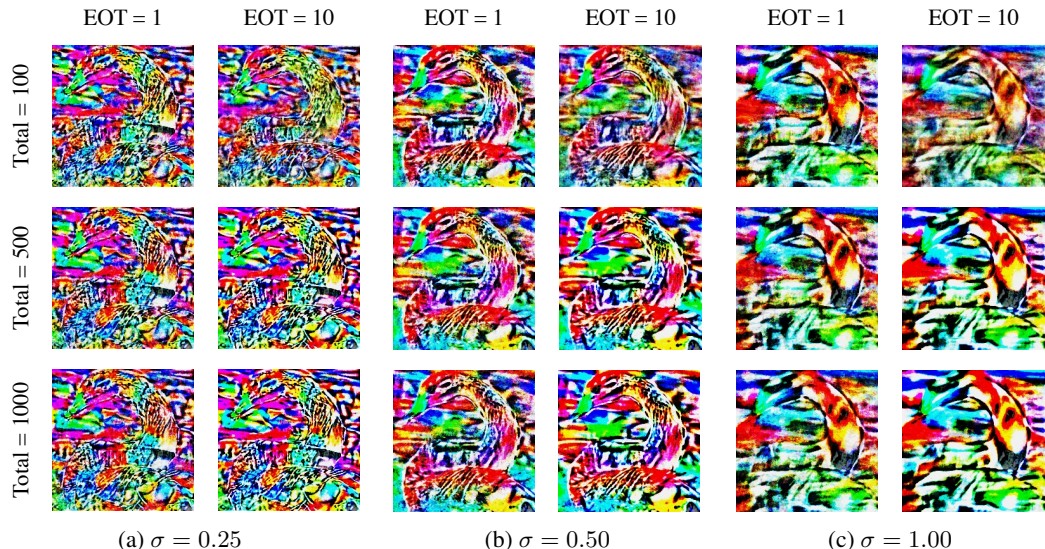

|  | EOT = 1 | EOT = 10 | EOT = 1 | EOT = 10 | EOT = 1 | EOT = 10 |

(a) $\sigma = 0.25$         (b) $\sigma = 0.50$         (c) $\sigma = 1.00$

Figure 10: Adversarial perturbation created by *untargeted* PGD attacks with and without EOT.

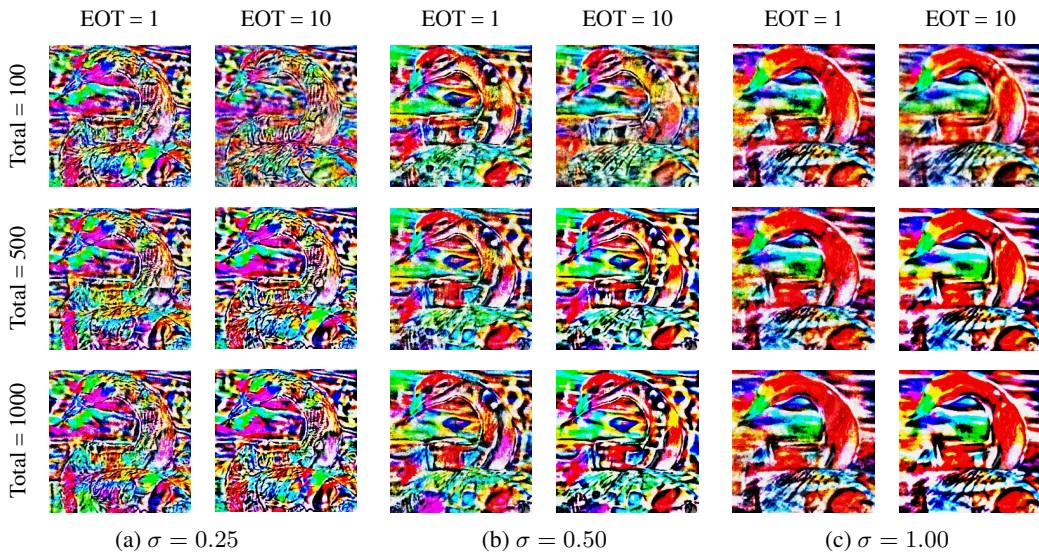

(a) $\sigma = 0.25$         (b) $\sigma = 0.50$         (c) $\sigma = 1.00$

Figure 11: Adversarial perturbation created by *targeted* PGD attacks with and without EOT.

provided by randomness. This observation corresponds to our theoretical model in Section 5 and empirical findings in Section 6.3.

### E.4   Additional Experiments on CIFAR10

In Table 2, we evaluate a few defenses on CIFAR10, including randomized activation pruning [8] and discontinuous activation [40]. Such defenses cannot defend against standard PGD attacks without applying EOT. This shows that our findings hold on small and large input spaces.

In this section, we add an experiment to show that our findings in Section 6.3 also hold on CIFAR10. To this end, we evaluate the randomized smoothing defense [6] on CIFAR10 with the more challenging targeted attack. Specifically, we run the standard PGD attack for 100 steps, with budget $\epsilon = 8/255$, step size $\alpha = 1/255$, target label 9, and EOT of $m = 20$ samples. The target models are ResNet-110 pre-trained on noisy data by Cohen et al. [6]. We evaluate four different noise levels $\sigma \in$

$\{0.12, 0.25, 0.50, 1.00\}$. For each noise level, we run the *same* attack on models before and after fine-tuning on data perturbed by such noise. The results are shown in Tables 3 and 4.

For all noise levels, fine-tuning models to obtain invariance improves the benign accuracy as expected. During this procedure, however, we can observe that the defense becomes less effective when the model recovers more invariance. In particular, the attack is nearly ineffective for $\sigma \in \{0.25, 0.50, 1.00\}$ when the model has low invariance, yet starts to work as the model recovers invariance. This observation is consistent with our findings on ImageNet in Section 6.3.

Table 3: Benign accuracy of models with low and high invariance to the defense's randomness.

|  | $\sigma = 0.12$ | $\sigma = 0.25$ | $\sigma = 0.50$ | $\sigma = 1.00$ |
|---|---|---|---|---|
| Before Fine-tuning (Low Invariance) | 23.4% | 14.7% | 12.3% | 10.1% |
| After Fine-tuning (High Invariance) | **83.6**% | **77.9**% | **71.1**% | **56.7**% |

Table 4: Attack success rate on models with low and high invariance to the defense's randomness.

|  | $\sigma = 0.12$ | $\sigma = 0.25$ | $\sigma = 0.50$ | $\sigma = 1.00$ |
|---|---|---|---|---|
| Before Fine-tuning (Low Invariance) | 52.1% | 1.1% | 0.0% | 0.0% |
| After Fine-tuning (High Invariance) | **63.1**% | **29.5**% | **18.1**% | **12.3**% |

# F   More Discussions

## F.1   Discussions about DiffPure [25]

In parallel to our work, DiffPure [25] adopts a complicated stochastic diffusion process to purify the input images. This defense belongs to an existing line of research that leverages generative models to pre-process input images and hence removing the potential adversarial perturbation [19, 32, 35]. In this section, we elaborate on the implications of our work for DiffPure.

***Firstly, DiffPure is the defense that our work expects to avoid.*** As we indicated in Section 1, a thorough evaluation of stochastic pre-processing defenses typically requires significant modeling and computational efforts. DiffPure is a new example of such defenses — it has a complicated solver of stochastic differential equations (SDE) and requires "high-end NVIDIA GPUs with 32 GB of memory[9]." Our initial experiment shows that it takes several hours to attack even one batch of 8 CIFAR10 images on an Nvidia RTX 2080 Ti GPU with 11 GB of memory, and we received an out-of-memory error when attempting ImageNet with batch size 1. Because of these complications and computational costs, fully understanding its robustness requires substantially more effort than a previous stochastic pre-processing defense BaRT [28].

Given this challenging arms race between attacks and defenses, our work provides empirical and theoretical evidence to show that stochastic pre-processing defenses are fundamentally flawed. They cannot provide inherent robustness (like that from adversarial training) to prevent the existence of adversarial examples. Hence, future attacks may break it. As a result of these findings, future research should look for new ways of using randomness, such as those discussed in Section 7.

***Secondly, DiffPure matches our theoretical model.*** DiffPure has two consecutive steps:

1. Forward SDE adds noise to the image to decrease invariance like Equation (5). The model becomes more robust because the input distribution is shifted.
2. Reverse SDE removes noise from the image to recover invariance like Equation (6). The model becomes less robust because the shifted input distribution is recovered.

These two steps are consistent with our characterization of stochastic pre-processing defenses in Section 5. While our work mainly focuses on trained invariance (through model fine-tuning), an auxiliary denoiser (like Reverse SDE) can achieve a similar notion of invariance. Hence, we expect our arguments about the robustness-invariance trade-off to hold here as well.

---

[9] https://github.com/NVlabs/DiffPure

***Finally, Our findings raise concerns with the way DiffPure claims to obtain robustness.*** The above discussion finds no evident difference between DiffPure and our model in Section 5. When the Reverse SDE is perfect, we should achieve full invariance in Equation (7) and expect no improved robustness — attacking the whole procedure is equivalent to attacking the original model (if non-differentiable and randomized components are handled correctly). Hence, our findings raise concerns with the way DiffPure claims to obtain robustness.

Driven by the above concerns, we carefully review DiffPure's evaluation and identify red flags:

1. They only used 100 PGD steps and 20–30 EOT samples in AutoAttack [7]. This setting is potentially inadequate based on our empirical results in Table 2. Even breaking a less complicated defense requires far more steps and samples.

2. Previous purification defenses cannot prevent adversarial examples on the manifold of their underlying generative model or denoiser [2]. However, DiffPure did not discuss this attack, i.e., whether it is possible to find an adversarial example of the diffusion model such that it remains adversarial (to the classifier) after the diffusion process. This strategy is different from its current evaluation, which attacks the whole pipeline with BPDA and EOT.

These red flags suggest that there is still room for improving DiffPure's evaluation.

***Summary.*** DiffPure matches our theoretical characterization of previous stochastic pre-processing defense. Thus, we expect our findings to hold here as well. Unfortunately, we cannot finish the evaluation of the above discussions due to their high computational costs. However, this challenge is exactly what our work aims to mitigate — we can identify concerns with the way robustness is achieved without needing to design adaptive attacks, and our findings have motivated us to identify red flags in their evaluation. We hope our work can increase the confidence of future research towards understanding the robustness of defenses sharing a similar assumption.

## F.2 Insights for Designing Attacks and Defenses Regarding Randomness

While systematic guidance for designing defenses (and their attacks) remains an open question, we attempt to summarize some critical insights for this direction as follows.

**Guidance for Attacks.**

1. Attackers aiming to evaluate defenses (i.e., not merely breaking them) should start with standard attacks before resorting to more involved attack strategies like EOT. This helps form a better understanding of the defense's fundamental weakness.

2. Stochastic pre-processors cannot provide inherent robustness, so an effective attack should exist. Although there has not been a systematic way to design or find such attacks, our work provides general guidelines to help with this task.

3. Stochastic pre-processors provide robustness by invariance, so attackers can examine the model invariance to check the room for improvements.

**Guidance for Defenses.**

1. The current use of randomness is not promising. Defenses should decouple robustness and invariance; otherwise, future attacks may break them.

2. Defenses should look for new ways of using randomness, such as those below or beyond the input space. Below-input randomness divides the input into orthogonal components, like modalities [43] and independent patches [18]. Beyond-input randomness routes the input to separate components, like non-transferable models [45].

3. Randomness should force the attack to target all possible (independent) subproblems, where the model performs well on each (independent and) non-transferable subproblem. In this case, defenses can decouple robustness and invariance, hence avoiding the pitfall of previous randomized defenses.

4. Randomness alone does not provide robustness. Defenses must combine randomness with other inherently robust concepts to improve robustness.

### F.3 Limitations and Potential Negative Societal Impacts

Finally, we discuss the limitations and potential negative societal impacts of this work.

**Limitations.** This paper mainly focuses on stochastic *pre-processing* defenses, thus we cannot comment on the effectiveness of stochastic defenses that are not based on input transformations. However, we do evaluate a few such defenses in Table 2 and observe similar results for our own interests, such as randomized activation pruning [8] and discontinued activation [40]. Given this observation, we believe our findings on stochastic pre-processing defenses are potentially generalizable to all stochastic defenses. We leave this exploration to future work.

Due to the limitation of computing resources, we are unable to evaluate the full BaRT [28] defense on ImageNet. To mitigate this problem, we evaluate a subset of BaRT on the smaller ImageNette dataset. Since the primary objective of this work is to study the limitations of such defenses but not to break them, we believe the limitations that we observe on a subset of BaRT are reasonably generalizable to the full set of BaRT. Other work studying this defense made a similar choice [34].

We are unable to evaluate the parallel defense DiffPure [25] due to its significantly high computational requirements. Given this limitation, we provide a thorough discussion in Appendix F.1 and explain that DiffPure is consistent with our model, hence we expect our findings to hold here as well.

**Potential Negative Societal Impacts.** This paper investigates the limitations of stochastic pre-processing defenses against adversarial examples. While the publication of this research may be used by attackers to create stronger attacks, we argue such considerations are out-weighted by the benefits of enabling defenders to understand the weaknesses of existing defenses. Moreover, our evaluation mainly involves existing attacks and previously broken defenses, thus we do not observe novel negative societal impacts. Our main objective is to uncover the fundamental weaknesses of such defenses, both empirically and theoretically, thereby raising the awareness of how to design proper stochastic defenses that avoid inadvertently weak evaluations and overestimated security.