# OpenReview forum: "On the Limitations of Stochastic Pre-processing Defenses"
_NeurIPS.cc/2022/Conference — NeurIPS 2022 Accept_

### Official Review · Reviewer_wZxu · 2022-06-27

**Rating:** 6
**Confidence:** 4
**Soundness:** 3 good
**Presentation:** 3 good
**Contribution:** 2 fair

**Summary:**

This work investigates stochastic pre-processing defenses both empirically and theoretically. Specifically, the authors first revisit previous stochastic pre-processing defenses and explain why such defenses are broken. Then, the authors study recent stochastic defenses that exhibit more randomness and show that they also face key limitations.

－－post rebuttal－－
Thank the author for providing detailed rebuttals. My major concerns are addressed. I raise my score and tend to accept this paper.

**Questions:**

I expected systematic guidance to design attacks and defenses regarding randomness from this paper. It will bring more value to the community if the author can provide one. The one provided in Discussion section is a good starting point.

I also encourage the authors to provide the expected experiments listed in weaknesses.


**Limitations:**

This work provides an analysis of the existing techniques, namely, stochastic defense. The new technique is the focus of this paper. The only limitation of this work is its novelty.

**Strengths And Weaknesses:**

Strengths:
1. This work focuses on an interesting topic, namely, the effectiveness of stochastic defenses.
2. In general, the paper is well organized.

Weaknesses:
1. My major concern is the novelty of the findings. The finding that applying EOT is only beneficial when the defense is sufficiently randomized is trivial to me. The other finding is that a trade-off between the model’s robustness provided by the defense and its invariance to the applied defense, which is also expected. When the model becomes invariant to the applied randomness, the function modeled by a model is more ‘deterministic’. The randomness is expected to bring less robustness to the model.

2. Some claims made in this paper are inappropriate, to my knowledge, e.g., in Line 46-47. The work [2] shows that EOT can be used to break stochastic defenses. However, the necessity of EOT is not claimed since they mainly aim to show that stochastic defenses can be easily broken with EOT.

3. Besides the success rate, an analysis of created adversarial noises should be presented. What is the difference between the adversarial noises created by PGD(n*m) or PGD(n)+EOT(m)? Similarly, what is the difference between the adversarial noises created on models with different degrees of randomness? The visualization of the noises might reveal more interesting conclusions.

4. It is great that the main experiments are conducted on ImageNet dataset. It will be better if the experiments on a smaller dataset, e.g.CIFAR10, can be provided. It is not clear if the experimental observation still holds when the input space is smaller.

---

> ### Author Response · Authors · 2022-08-02
> **Response to Reviewer wZxu 2/2**
>
> **Q2: Systematic guidance to design attacks and defenses regarding randomness.**
>
> Thank you for recognizing the value of our discussion section. We also discussed implications for future research in our response to the first reviewer’s Q1. Below we elaborate on the specific questions you proposed; we hope these discussions can sharpen our contributions to the community.
>
> First, we want to kindly note that systematic guidance for designing attacks and defenses remains an open question, as evident from the difficulty of designing general adaptive attacks and robust defenses [3]. Still, our work suggests critical insights in this direction.
>
> Guidance for attacks:
> 1. Attackers aiming to evaluate defenses (i.e., not merely breaking them) should start with standard attacks before resorting to more involved attack strategies like EOT. This helps form a better understanding of the defense’s fundamental weakness.
> 2. Randomized pre-processors cannot provide inherent robustness, so an effective attack should exist. Although there has not been a systematic way to design such attacks, our work provides general guidelines to help with this task.
> 3. Randomized pre-processors provide robustness by invariance, so attackers can examine the model invariance to check the room for improvements.
>
> Guidance for defenses:
> 1. The current use of randomness is not promising. Defenses should decouple robustness and invariance; otherwise, future attacks may break them.
> 2. Defenses should look for new ways of using randomness, such as those below or beyond the input space. Below-input randomness divides the input into orthogonal components, like modalities [4] and independent patches [5]. Beyond-input randomness routes the input to separate components, like non-transferable models [6].
> 3. Randomness should force the attack to target all possible (independent) subproblems, where the model performs well on each (independent and) non-transferable subproblem. In this case, defenses can decouple robustness and invariance, hence avoiding the pitfall of previous randomized defenses.
> 4. Randomness alone does not provide robustness. Defenses must combine randomness with other inherently robust concepts to improve robustness.
>
> [4] Defending Multimodal Fusion Models against Single-Source Adversaries. CVPR 2021.
>
> [5] Certified Robustness Against Physically-Realizable Patch Attacks via Randomized Cropping. ICML 2021 Workshop.
>
> [6] TRS: Transferability Reduced Ensemble via Promoting Gradient Diversity and Model Smoothness. NeurIPS 2021.
>
> **Q3: Claims about the necessity of EOT are inappropriate.**
>
> Thank you for pointing this out; we tried to be very careful about these claims. We have revised our claims to clarify that we do not intend to claim EOT as being unnecessary. Rather, we use this finding to show that adaptive attacks might overestimate the robustness of weak defenses (L51-53).
>
> Despite this, we kindly note that several adaptive attack papers have indeed claimed EOT as necessary (or at least standard). For example, AutoAttack [7] explicitly detects random components and enforces EOT, and other attacks [8] regard EOT as a "standard technique for computing gradients of models with randomized components." Such default settings could worsen the above overestimation of the defense robustness.
>
> [7] Reliable Evaluation of Adversarial Robustness with an Ensemble of Diverse Parameter-free Attacks. ICML 2020.
>
> [8] On Adaptive Attacks to Adversarial Example Defenses. NeurIPS 2020.
>
> **Q4: Visualization of adversarial noises.**
>
> Adversarial noises are smoother when created (1) with EOT or (2) on models with more randomness. This observation shows that (1) applying EOT and (2) making models invariant to more randomness both lead to more stable gradients (hence easier to attack and confirm our findings). Please find detailed results and discussions in the revised Appendix E.3.
>
> **Q5: Evaluation of CIFAR10.**
>
> We test randomized smoothing on CIFAR10, and our observations hold. The model becomes less robust to the same attack after fine-tuning, as shown in the table below. Please find detailed results in the revised Appendix E.4.
>
> | Attack Success Rates   | $\sigma=0.12$ | $\sigma=0.25$ | $\sigma=0.50$ | $\sigma=1.00$ |
> |-------------------:|:-------------:|:-------------:|:-------------:|:-------------:|
> | Before Fine-tuning |        52.1% |        1.1% |        0.0% |        0.0% |
> |  After Fine-tuning |        **63.1%** |        **29.5%** |        **18.1%** |        **12.3%** |

---

> > ### Comment · Reviewer_wZxu · 2022-08-09
> > **Post-rebuttal**
> >
> > Thank the author for providing detailed rebuttals. My major concerns are addressed. I raise my score and tend to accept this paper.

---

> > > ### Author Response · Authors · 2022-08-09
> > > **Thank you!**
> > >
> > > We appreciate the reviewer's detailed review and positive comments. We will incorporate these discussions and additional experimental results into the main paper and provide more details in the appendix.

---

> ### Author Response · Authors · 2022-08-02
> **Response to Reviewer wZxu 1/2**
>
> Thank you for your insightful comments. Our detailed response to your major concerns is below.
>
> **Q1: The findings are expected.**
>
> While our findings are expected, they have not been well recognized by the research community (detailed below). This problem is under-explored and lacks theoretical support. *Our main contribution lies in bridging this gap with a theoretical underpinning for the "expected ineffectiveness" of stochastic pre-processing defenses.* It provides key insights that future research should (at least try to) abandon this assumption.
>
> Below we highlight the significance of our contributions.
>
> **1. The "expected" findings have not been well recognized by the research community.**
>
> The research community continues improving defenses through more complicated transformations. For example, there is a new stochastic pre-processing defense DiffPure [1] at ICML this year (published after our submission). This defense has a complicated solver of stochastic differential equations (SDE) and requires high-end GPUs with 32 GB of memory [2]. Our initial experiment shows that it takes several hours to attack even one batch of 8 CIFAR10 images on an Nvidia RTX 2080 Ti GPU with 11 GB of memory, and we received an out-of-memory error when attempting ImageNet with batch size 1. Because of these complications and computational costs, fully understanding DiffPure’s robustness requires substantially more effort than a previous stochastic pre-processing defense BaRT, which was only broken after 3 years of its publication.
>
> [1] Diffusion Models for Adversarial Purification. ICML 2022.
>
> [2] https://github.com/NVlabs/DiffPure
>
> **2. Our theoretical results have the potential to include newer defenses like DiffPure.**
>
> DiffPure has two consecutive steps:
> 1. Forward SDE adds noise to the image to decrease invariance. The model becomes more robust (Eq. 5) due to shifted input distribution.
> 2. Reverse SDE removes noise from the image to recover invariance. The model becomes less robust (Eq. 6) due to recovered input distribution.
>
> These two steps are consistent with our characterization of stochastic pre-processing defenses in Section 5. While our submission mainly focused on trained invariance (through model fine-tuning), an auxiliary denoiser (like Reverse SDE) can achieve a similar notion of invariance. Hence, we expect our arguments about the robustness-invariance trade-off to hold here as well.
>
> **3. Our findings raise concerns with the way DiffPure claims to obtain robustness.**
>
> The above discussion finds no evident difference between DiffPure and our model. When the Reverse SDE is perfect, we should achieve full invariance (Eq. 7) and expect no improved robustness — attacking the whole procedure is equivalent to attacking the original model (if non-differentiable and randomized components are handled correctly). Hence, our findings raise concerns with the way DiffPure claims to obtain robustness.
>
> Driven by these concerns, we carefully reviewed DiffPure’s evaluation and identified red flags.
> 1. They only used 100 PGD steps and 20 EOT samples in AutoAttack. This setting is potentially inadequate based on our empirical results (e.g., Tab. 2). Even breaking a less complicated defense requires far more steps and samples.
> 2. Previous purification defenses cannot prevent adversarial examples on the manifold of their underlying generative model or denoiser [3]. However, DiffPure did not discuss this attack, i.e., whether it is possible to find an adversarial example of the diffusion model such that it remains adversarial (to the classifier) after the diffusion process. This strategy is different from attacking the whole pipeline with BPDA and EOT.
>
> These red flags suggest that there is still room for improving DiffPure’s evaluation.
>
> [3] Obfuscated Gradients Give a False Sense of Security: Circumventing Defenses to Adversarial Examples. ICML 2018.
>
> **4. Our findings help to mitigate the challenges of robustness evaluation.**
>
> We cannot finish the evaluation of the above discussions within the short rebuttal period, mainly due to the complicated nature of stochastic pre-processing defenses and their high computational costs. However, this challenge is exactly what our work aims to mitigate — we can identify concerns with the way robustness is achieved without needing to design adaptive attacks, and our findings have motivated us to identify several red flags in their robustness evaluation.
>
> Lastly, while the findings are expected, our theoretical results increase the confidence of future research towards understanding the robustness of defenses relying on a similar assumption. This problem is under-explored and lacks theoretical support. Our work bridges this gap and helps to mitigate the arms race between attacks and defenses.

---

> ### Author Response · Authors · 2022-08-08
> **Looking forward to further discussions.**
>
> We would like to thank you again for your detailed review and insightful comments. We hope our responses can adequately address your concerns and encourage you to reconsider the score. We sincerely look forward to further discussions if you have any questions.

---

### Official Review · Reviewer_G9xL · 2022-07-10

**Rating:** 7
**Confidence:** 4
**Soundness:** 4 excellent
**Presentation:** 4 excellent
**Contribution:** 3 good

**Summary:**

"On the Limitations of Stochastic Pre-processing Defenses" discusses strategies for the adequate evaluation of adversarial attacks against systems defended by stochastic components. Previous work generally considers expectation-over-transformations to be necessary to attack these defenses, but the submission shows that merely running standard defenses with larger step budgets can often be sufficient in overcoming these defenses. The submissions supplements this finding with further analysis on the interactions of robustness and invariance with the stochastic nature of these defenses.

**Questions:**

As described above, how would a stochastic defense based on diffusion processes be categorized? Do the findings of this work for older defenses also hold for these newer stochastic defenses?

**Limitations:**

n.a.

**Strengths And Weaknesses:**

I like the clarity of writing and presentation of the findings in this work. The grouping of stochastic defenses into defenses with and without meaningful stochasticity is nicely illustrated and exemplified.

The experimental evaluation underpins these findings for two defenses, barrage-of-transformations and randomized smoothing. Overall I don't have a lot of say here, I liked the evaluations done by the authors and find the results interesting. The overall topic is not groundbreaking, but reinvestigating the properties of stochastic preprocessing is, in my opinion, illuminating for the wider community.

The only minor weakness I see is that the evaluations center on two classic stochastic defenses. Recently there has been a lot of renewed interest in stochastic defenses, especially based on diffusion processes (For example Nie et al, "Diffusion Models for Adversarial Purification
" at ICML this year). I think the impact of this submission could be improved by also considering one such "newer" attempted adversarial defense, where would this defense fall in the categorization of stochastic defenses proposed in this work?

---

> ### Author Response · Authors · 2022-08-02
> **Response to Reviewer G9xL**
>
> Thank you for your positive and insightful comments.
>
> The framework of DiffPure [1] is consistent with our model. Hence, we expect our arguments about the robustness-invariance trade-off to hold here as well: Applying the diffusion preprocessor makes the model invariant to the noise and less robust. We also identify red flags in their robustness evaluation based on our findings. Please note that DiffPure was published after our submission. Our detailed response is below.
>
> **1. DiffPure is the defense that our work expects to avoid.**
>
> As we indicated in the introduction, a thorough evaluation of stochastic pre-processing defenses typically requires significant modeling and computational efforts. DiffPure is a new example of such defenses — it has a complicated solver of stochastic differential equations (SDE) and requires high-end GPUs with 32 GB of memory [2]. Our initial experiment shows that it takes several hours to attack even one batch of 8 CIFAR10 images on an Nvidia RTX 2080 Ti GPU with 11 GB of memory, and we received an out-of-memory error when attempting ImageNet with batch size 1. Because of these complications and computational costs, fully understanding its robustness requires substantially more effort than a previous stochastic pre-processing defense BaRT, which was only broken after 3 years of its publication.
>
> Given this challenging arms race between attacks and defenses, our work provides empirical and theoretical evidence to show that stochastic pre-processing defenses are fundamentally flawed. They cannot provide inherent robustness (like that from adversarial training) to prevent the existence of adversarial examples. Hence, future attacks may break it. As a result of these findings, future research should look for new ways of using randomness, such as those discussed in L322-328.
>
> [1] Diffusion Models for Adversarial Purification. ICML 2022.
>
> [2] https://github.com/NVlabs/DiffPure
>
> **2. DiffPure matches our theoretical model.**
>
> DiffPure has two consecutive steps:
> 1. Forward SDE adds noise to the image to decrease invariance. The model becomes more robust (Eq. 5) due to shifted input distribution.
> 2. Reverse SDE removes noise from the image to recover invariance. The model becomes less robust (Eq. 6) due to recovered input distribution.
>
> These two steps are consistent with our characterization of stochastic pre-processing defenses in Section 5. While our submission mainly focused on trained invariance (through model fine-tuning), an auxiliary denoiser (like Reverse SDE) can achieve a similar notion of invariance. Hence, we expect our arguments about the robustness-invariance trade-off to hold here as well.
>
> **3. Our findings raise concerns with the way DiffPure claims to obtain robustness.**
>
> The above discussion finds no evident difference between DiffPure and our model. When the Reverse SDE is perfect, we should achieve full invariance (Eq. 7) and expect no improved robustness — attacking the whole procedure is equivalent to attacking the original model (if non-differentiable and randomized components are handled correctly). Hence, our findings raise concerns with the way DiffPure claims to obtain robustness.
>
> **4. DiffPure’s robustness evaluation has red flags.**
>
> Driven by the above concerns, we carefully reviewed DiffPure’s evaluation and identified red flags.
> 1. They only used 100 PGD steps and 20 EOT samples in AutoAttack. This setting is potentially inadequate based on our empirical results (e.g., Tab. 2). Even breaking a less complicated defense requires far more steps and samples.
> 2. Previous purification defenses cannot prevent adversarial examples on the manifold of their underlying generative model or denoiser [3]. However, DiffPure did not discuss this attack, i.e., whether it is possible to find an adversarial example of the diffusion model such that it remains adversarial (to the classifier) after the diffusion process. This strategy is different from attacking the whole pipeline with BPDA and EOT.
>
> These red flags suggest that there is still room for improving DiffPure’s evaluation.
>
> [3] Obfuscated Gradients Give a False Sense of Security: Circumventing Defenses to Adversarial Examples. ICML 2018.
>
> **5. Summary.**
>
> DiffPure matches our theoretical characterization of previous stochastic pre-processing defense. Thus, we expect our findings to hold here as well. Unfortunately, we cannot finish the evaluation of the above discussions within the rebuttal period due to their high computational costs. If we are able to get results, we will add them during the discussion period.
>
> However, this challenge is exactly what our work aims to mitigate — we can identify concerns with the way robustness is achieved without needing to design adaptive attacks, and our findings have motivated us to identify red flags in their evaluation. We hope our work can increase the confidence of future research towards understanding the robustness of defenses sharing a similar assumption.

---

> > ### Comment · Reviewer_G9xL · 2022-08-04
> > **About DiffPure**
> >
> > Thank you for the additional explanations and sorry for springing this new information on you during the rebuttal phase. Including the provided discussion about DiffPure in a camera-ready version /  revision would already be meaningful to me, even without additional experiments and raise my score of this submission further.
> >
> > Of course, additionally verifying that the proposed analysis correctly predicts that DiffPure is vulnerable, would be even more impactful, but, I agree, this is tough to engineer within a short amount of time given the computational requirements of the defense.

---

> > > ### Author Response · Authors · 2022-08-04
> > > **Thank you!**
> > >
> > > We appreciate you bringing up the insightful discussion as it is an excellent fit for our motivation. We will incorporate these discussions to the main paper and provide more details in the appendix. We hope that this adequately addresses your concerns to raise the score further.

---

> > > > ### Comment · Reviewer_G9xL · 2022-08-05
> > > > **.**
> > > >
> > > > Just for the record, I've done so.

---

### Official Review · Reviewer_SAKU · 2022-07-15

**Rating:** 6
**Confidence:** 4
**Soundness:** 3 good
**Presentation:** 3 good
**Contribution:** 2 fair

**Summary:**

This paper contains three main contributions. First, this paper shows that using EoT ($m>1$) is not always necessary to break randomized defenses, i.e., it does not necessarily save the number of gradient computations. Second, it shows a theoretical result in a toyish setting that the robustness of randomized models is at odds with their utility (accuracy). This is not an impossibility result but rather, the authors propose some settings where such a trade-off exists. Lastly, the paper supports the second the previously mentioned theoretical results with empirical ones on two types of defenses (BaRT and randomized smoothing).

**Questions:**

1. What about the robustness-invariance trade-off in the targeted setting? The invariance is needed for accuracy when it represents $F_\theta(x) = y$, i.e., "prediction has to be correct after the random transform." However, the defender needs *no* invariance for the wrong class, correct? We don't care, almost in all cases, that $F_\theta(x) = F(x)$ when they are both wrong. The defense can surely use this to its advantage. I expect that the robustness-invariance trade-off is far less important, if it exists at all, in the targeted setting.
2. In Figure 4a and 4c, we can see that the red and the green solid lines start to diverge at large $\sigma$. This means that the trade-off somewhat disappears here, i.e.,  you gain more robustness while the accuracy does not go down. Why does this happen on randomized smoothing and not BaRT? Why is this the case? Could it be that the attack is sub-optimal with large $\sigma$? It might be good to show that the attack does indeed converge by no large $m$ and $k$ could possibly increase the attack success rate further.
3. Do all attacks in the paper use AutoPGD? If so, did the authors take into account the observation that EoT is unnecessary could be, at least partially, an effect of the adaptive step size from AutoPGD?

**Limitations:**

This is adequately discussed in Appendix F.

**Strengths And Weaknesses:**

# Strengths
1. The paper addresses an interesting yet under-explored problem. Rigorous studies are lacking on randomized models/defenses.
2. While the setup for the theoretical part is toyish, the results are comprehensive. I would have missed the more interesting results, in my opinion, in the appendix if I didn't check out Appendix C.2. The paper might benefit from bringing up or summarizing some of these results in the main paper.
3. Experiment setup seems mostly rigorous and up to date with recent works.
4. The presentation is good. The main messages are all easy to understand, and the results mostly support the claims well.

# Weaknesses
1. The main and only major weakness is that the overall contribution is slightly limited (justified in a. and b. below). Personally, I find the two main observations in this paper interesting. However, the paper seems to miss the final part, i.e., I find the following questions unanswered: What's next? What does this mean to future research? How do we improve defenses? Should we abandon randomized defenses? What are concrete settings that this defense works (with some supporting results)?
  a. *Theory*: As mentioned in the summary, there is no guarantee that the trade-off between utility, i.e., invariance, and robustness must hold. The results do cover a reasonable set of settings used by the proposed defenses, but none really exclude the possibility of using other forms of randomness or aggregation methods that may not conform to this trade-off or in general achieve a better trade-off.
  b. *Empirical*: I don't believe that the empirical contribution quite makes up for the limitation in the theoretical part. Additional experiments on more models and defenses might be good. Surely, all defenses are not made equal and so they should not be on the same trade-off curve. Could the authors make a stronger claim about which defense might look more promising? Or maybe a heuristic to properly choose $m$ and $k$ without just trying out a lot of them?

Note that I'm not set in stone about this. The authors and the other reviewers may convince me otherwise.

# Minor Weaknesses
I think of these as suggestions or room for improvement rather than things I penalize this paper for.
1. I believe that Pinot et al. [2021] (https://arxiv.org/abs/2102.10875) might be related to this work, at least remotely. This work is not peer-reviewed as far as I can find so I'm not imposing any penalty. However, I'd recommend citing and discussing their result as well since they also show many solid theoretical results on randomized defense.
2. Maybe the comparison between the benefits of the number of PGD steps ($m$) and the number of EoT samples ($k$) can be put in the perspective of stochastic gradient descent. For instance, from the central limit theorem, we know that variance decreases by $1/k$ and $1/\sqrt{k}$ for standard deviation. We also know that the variance of the gradient estimates affects the convergence rate of SGD (see this [paper](https://cpn-us-w2.wpmucdn.com/sites.gatech.edu/dist/f/330/files/2016/02/NonconvexSA-Revision-9-25-13.pdf)). This might allow one to predict, in theory, the optimal value of $k$ and $m$ given $\sigma$ and $k \times m$.
3. Presentation of Figure 3 and 4: I don't think the epoch views are necessary. I would also recommend plotting the attack success rate or robust accuracy vs clean accuracy to show the trade-off better.

# Very Trivial Weaknesses
1. Reference [10] and [11] are duplicates.
2. It's a bit difficult to compare a fixed number of gradient computation ($k \times m$) in Figure 1 and 2. It just takes another little step to multiply the numbers. I don't have a good suggestion for this without significantly changing the figure (e.g., x-axis is just $k \times m$).

---

> ### Author Response · Authors · 2022-08-02
> **Response to Reviewer SAKU 2/2**
>
> **Q4: The robustness-invariance trade-off may be less important in the targeted setting.**
>
> The reviewer is correct that the defender needs no invariance for the wrong class, and it is hard for our theoretical model (in the binary regime) to cover the multi-class targeted setting. However, our experiments already covered the class-dependent invariance, as our training loss does not penalize the inconsistent wrong prediction $F_\theta(x)=F(x)\neq y$.
>
> Figures 4a and 4c show that the robustness-invariance trade-off does exist in the targeted setting and is more critical than in the untargeted setting. While untargeted attacks obtain ~20% higher success rates after model fine-tuning, the success rates of targeted attacks increase from ~0% to 50-80%. This is because we still need to preserve invariance (or utility) in the targeted setting, and the low invariance is more challenging for targeted attacks. As a result, model fine-tuning makes targeted attacks more effective than untargeted attacks.
>
> **Q5: The trade-off somewhat disappears in Figures 4a and 4c.**
>
> The reviewer is correct that the attack is suboptimal for large noise. However, we choose this setting *on purpose* to compare the robustness under the *same* attack before and after fine-tuning. Under this setting, we can observe that the same attack (regardless of its strength) that hardly works for the defense (before fine-tuning & low invariance) now becomes more effective (after fine-tuning & high invariance). We can surely run each attack for more iterations and samples, but the current setting suffices to show that the defense provides robustness by explicitly reducing invariance.
>
> **Q6: EOT being unnecessary could be an effect of the adaptive step size from AutoPGD.**
>
> We are also cautious about this setting. For all motivation experiments in Section 4 and main experiments in Section 6.2, we only use standard PGD with constant step size and no random restarts.

---

> ### Author Response · Authors · 2022-08-02
> **Response to Reviewer SAKU 1/2**
>
> Thank you for your detailed review and insightful comments. We will address the minor weaknesses and are glad to continue the discussion. Our detailed response to your major concerns is below.
>
> **Q1: The paper seems to miss the final part.**
>
> We briefly covered a few questions in the discussion section, such as how defenses should utilize randomness and the implications for adaptive attackers. Below we elaborate on the specific questions you proposed; we hope these discussions can sharpen our contributions.
>
> > *What's next? What does this mean to future research?*
>
> Our work suggests that future defenses should decouple robustness and invariance; that is, avoid providing robustness by introducing variance to the added randomness. Otherwise, defenses that shift the input distribution will result in errors, and the observed "robustness" is only a result of these errors. These findings imply that future research should (at least try to) abandon this assumption.
>
> This implication is crucial as the research community continues improving defenses through more complicated transformations. For example, there is a new stochastic pre-processing defense DiffPure [1] at ICML this year (published after our submission). This defense has a complicated solver of stochastic differential equations (SDE) and requires high-end GPUs with 32 GB of memory [2]. Our initial experiment shows that it takes several hours to attack even one batch of 8 CIFAR10 images on an Nvidia RTX 2080 Ti GPU with 11 GB of memory, and we received an out-of-memory error on ImageNet with batch size 1. Because of such complications and high computational costs, fully understanding DiffPure’s robustness requires substantially more effort than a previous stochastic pre-processing defense BaRT, which was only broken after 3 years of its publication.
>
> [1] Diffusion Models for Adversarial Purification. ICML 2022.
>
> [2] https://github.com/NVlabs/DiffPure
>
> > *How do we improve defenses? Should we abandon randomized defenses?*
>
> We should not abandon randomized defenses but utilize randomness in new ways. One promising approach is dividing the problem into orthogonal subproblems. For example, some speech problems (such as keyword spotting) are inherently divisible in the spectrum space, and vision tasks are divisible by introducing different modalities [3], independency [4], or orthogonality [5]. In such cases, randomization forces the attack to target all possible (independent) subproblems, where the model performs well on each (independent and) non-transferable subproblem. As a result, defenses can decouple robustness and invariance, hence avoiding the pitfall of previous randomized defenses.
>
> [3] Defending Multimodal Fusion Models against Single-Source Adversaries. CVPR 2021.
>
> [4] Certified Robustness Against Physically-Realizable Patch Attacks via Randomized Cropping. ICML 2021 Workshop.
>
> [5] TRS: Transferability Reduced Ensemble via Promoting Gradient Diversity and Model Smoothness. NeurIPS 2021.
>
> > *What are concrete settings that this defense works?*
>
> Randomized defenses make the attack harder in the black-box setting (L315-321). However, we cannot find evidence that stochastic pre-processing defenses work in the white-box setting. Other forms of randomness discussed above are more promising. The only exception is randomized smoothing, which remains an effective tool to certify the inherent robustness of a given decision.
>
> **Q2: Theory cannot exclude other forms of randomness.**
>
> We clarify that we are not making a statement about stochastic defenses in general. Instead, we focus on a popular subclass that leverages randomness to affect the model’s invariance. Other forms of randomness that our theory cannot exclude are exactly what we advocate for future research, like those discussed in Q1.
>
> However, we can use our results to argue about newer defenses. For example, DiffPure has two consecutive steps:
> 1. Forward SDE adds noise to the image to decrease invariance.
> 2. Reverse SDE removes noise from the image to recover invariance.
>
> These two steps correspond to our characterization of stochastic pre-processing defenses in Section 5. While our submission mainly focused on trained invariance (through model fine-tuning), an auxiliary denoiser (like Reverse SDE) can achieve a similar notion of invariance. Hence, we expect our arguments about the robustness-invariance trade-off to hold here as well.
>
> **Q3: A stronger claim about which defenses might look more promising.**
>
> We clarify that the main purpose of our evaluation is *not* to find defenses with a better robustness-invariance trade-off, but to show that they do have this trade-off. Such a trade-off implies that the defense uses randomness to affect the invariance and thus cannot provide inherent robustness (like that from adversarial training). A stronger (negative) claim based on our empirical results would be: Any defenses facing the robustness-invariance trade-off are not promising.

---

> ### Author Response · Authors · 2022-08-08
> **Looking forward to further discussions.**
>
> We would like to thank you again for your detailed review and insightful comments. We hope our responses can adequately address your concerns and encourage you to reconsider the score. We sincerely look forward to further discussions if you have any questions.

---

> ### Comment · Reviewer_SAKU · 2022-08-09
> **Comments on Authors' Response**
>
> First, I would like to apologize for the delay in responding, and I thank the authors for their thoughtful responses.
>
> After having read the other reviews and the responses, I believe that my main and only concern still stands. However, the other questions I posted are reasonably addressed, or at least, acknowledged. I hope that the authors incorporate the explanations they provided into the final version of the paper. Overall, I believe that this paper meets the standard of NeurIPS publications and would benefit the overall community and other researchers. Therefore, I decided to increase my score from 5 to 6.

---

> > ### Author Response · Authors · 2022-08-09
> > **Thank you!**
> >
> > We appreciate the reviewer's detailed review and positive comments. We will incorporate these discussions into the main paper and provide more details in the appendix.

---

### Author Response · Authors · 2022-08-05
**General Response**

Dear Program Chairs, Area Chairs, and Reviewers,

We would like to thank all reviewers for their time and very much appreciate their assessment of our work as *"interesting yet under-explored"* (SAKU, G9xL, wZxu) with *"rigorous and comprehensive results"* (SAKU) that are *"illuminating for the wider community"* (G9xL). The concerns are mainly focused on "the findings are expected" and "the final part questions" of this paper. We summarize and highlight our responses to these concerns below.

**Q1: Findings are expected.**

Reviewer wZxu raises a concern that our findings are "expected." While this concern is valid, we want to kindly note that such "expected findings" have not been well recognized by the research community (Q2). In fact, it is precisely this *gap* between the community's effort and the reviewer's correct understanding (of "expected" findings) that our work aims to bridge. Hence, our main contribution is to *provide a theoretical underpinning for the "expected ineffectiveness" of stochastic pre-processing defenses.* This is a novel insight that has not been rigorously studied.

**Q2: DiffPure at ICML this year.**

Reviewer G9xL brings up an insightful [discussion](https://openreview.net/forum?id=P_eBjUlzlV&noteId=tr7DOUtqQgX) about DifffPure, a new stochastic pre-processing defense published at ICML this year (after our submission). This new defense is strong evidence showing that the community continues improving defenses through more complicated randomized transformations. Fortunately, this new defense fits our model, so we can use our findings to identify concerns with the way its robustness is achieved without needing to design adaptive attacks. We hope this discussion can help solidify the novelty and impact of our work from a different perspective.

**Q3: The final part questions.**

While reviewers endorse our discussion section, they expect to see more implications for future research (SAKU) and systematic guidance to design attacks and defenses (wZxu). We understand that our discussion section was not as comprehensive as we hoped, and we very much appreciate for bringing up these insightful questions. We have provided an in-depth discussion of these questions in our response to Reviewers [SAKU](https://openreview.net/forum?id=P_eBjUlzlV&noteId=i1vRufZ61gt) and [wZxu](https://openreview.net/forum?id=P_eBjUlzlV&noteId=63Cry6fZp8o).

We hope our responses can adequately address the reviewers' concerns and encourage them to reconsider their scores. We sincerely look forward to further discussions with the reviewers if they have any questions.

Best wishes,

Authors of Submission 3796

---

### Meta-Review · Area_Chair_vjFS · 2022-08-28

**Recommendation:** Accept
**Confidence:** Less certain

**Metareview:**

This paper considers the effectiveness of stochastic preprocessing methods at achieving adversarial robustness. It shows empirically that the common Expectation of Transformations attack is not necessary to break many such defenses, as these defenses are vulnerable to standard PGD attacks when the amount of randomization is small. In a specific setup, the authors prove a trade-off between the utility and robustness of randomization defenses, and demonstrate on real data sets that such a trade-off exists for two randomized defenses (Barrage of Random Transforms and randomized smoothing). Although there is concern about the lack of clear impact on the development of future defense schemes, the reviewers found the message and empirical results of the paper to be illuminating.

**Award:**

No

---

### Decision · Program_Chairs · 2022-09-14

Accept